



# Mean Transit Times in Headwater Catchments: Insights from the Otway Ranges, Australia

## William Howcroft[1], Ian Cartwright[1, 2] and Uwe Morgenstern[3]

[1]School of Earth, Atmosphere and Environment (SEAE), 9 Rainforest Walk (Building 28),

Monash University, Clayton Campus, Clayton, VIC, 3800, Australia.

[2]National Centre for Groundwater Research and Training, GPO Box 2100, Flinders University,

Adelaide, SA, 5001, Australia.

[3]GNS Science, 1 Fairway Drive, Avalon, PO Box 368, Lower Hutt 5040, New Zealand.

*Correspondence to*: william.howcroft@monash.edu

## Abstract

Understanding the timescales of water flow through catchments and the origins of stream water at different flow conditions is critical for understanding catchment behaviour and managing water resources. Here, tritium ($^3H$) activities, major ion geochemistry and

discharge data were used in conjunction with Lumped Parameter Models (LPMs) to investigate mean transit times (MTTs) and the stores of water in six headwater catchments of the Otway Ranges in southeast Australia. $^3H$ activities of stream water ranged from 0.20 to 2.14 TU, which are far lower than those of modern local rainfall (2.4 to 3.2 TU). The $^3H$ activities of the stream water are lowest during the low summer flows and increase with

stream discharge. Calculated MTTs vary from approximately 7 to 234 years which, in many cases, exceed those reported for river systems globally. The MTT estimates, however, are subject to a number of uncertainties, including, uncertainties in the most appropriate LPM to use, aggregation errors, and uncertainty in the modern and bomb-pulse $^3H$ activity of rainfall. These uncertainties locally result in uncertainties in MTTs of several years;

however, they do not change the overall conclusions that the water in these streams has MTTs of several years to decades. There is discharge threshold of approximately $10^4$ m$^3$ day$^{-1}$ in all catchments above which $^3H$ activities do not increase appreciably above ~2.0 TU. The MTT of this $^3H$ activity is approximately ten years, which implies that changes within the





catchments, including drought, deforestation, land use and/or bush fire, would not be

realised within the streams for at least a decade. A positive correlation exists between $^3$H

activities and nitrate and sulphate concentrations within several of the catchments, which

suggests that anthropogenic activities have increasingly impacted water quality at these

locations over time.

## 1. Introduction

The timescales over which precipitation is transmitted from a recharge area through an

aquifer to where it discharges into rivers or springs (the transit time) is of inherent interest

to resource managers. Changes to the land use within a catchment, including deforestation

and/or agricultural development together with bushfires, drought, deforestation or

contaminant loading, can affect both the quality and the quantity of river flows.

Documenting the MTTs allows the timescales over which such changes may affect the

streams to be assessed. In recent years, there has been considerable research addressing

catchment transit times, for example as reviewed by McGuire and McDonnell (2006) and

McDonnell et al. (2010). Much of this research has focussed on understanding transit times

within upland (headwater) catchments (e.g. Mueller et al., 2013; Stockinger et al., 2014;

Cartwright and Morgenstern, 2015, 2016a).

Headwater streams are important for a variety of reasons: they commonly support diverse

ecosystems, provide unique recreational opportunities and, in many catchments, contribute

a significant proportion of the total river discharge (Freeman et al., 2007). Headwater

streams also differ from lowland rivers in terms of their potential water inputs. Unlike

lowland rivers, which typically receive inflows from regional groundwater and near-river

floodplain sediments, the source(s) of water within headwater streams is far less well

understood.

Headwater streams are commonly developed at elevations well above those of the regional

water tables and/or are seated upon relatively impermeable bedrock. Yet, such streams

continue to flow, even during prolonged dry periods. There are several potential water





stores that could contribute to stream flow, including the soil zone, weathered or fractured

basement rocks, and/or perched aquifers at the soil-bedrock interface. The relative

contribution of such stores to total stream flow has been examined for some decades now

(e.g. Sklash and Farvolden, 1979; Kennedy et al., 1986; Swistock et al., 1989; Bazemore et

al., 1994; Fenicia et al., 2006; and Jensco and McGlynn, 2011). However, the transit times

of such stores are less well understood. There are a growing number of estimates of transit

times in headwater catchments that range from a few months (e.g. Soulsby et al., 2000;

Stewart and Fahey, 2010; Duvert et al., 2016) to several years (Atkinson, 2014; Cartwright

and Morgenstern, 2015, 2016a). However, in many headwater catchments, the range of

transit times is not well known, nor are the catchment attributes that control the transit

times.

### 1.1. Estimation of Mean Transit Times (MTTs)

MTTs can be estimated from numerical groundwater models. However, the hydraulic

parameters used in such models are seldom known with great certainty and vary spatially,

which can lead to unrealistic estimates of MTTs. More frequently, MTTs are estimated using

geochemical tracers. These tracers include: stable (O, H) isotopes and major ion

concentrations that vary seasonally in rainfall, radioactive isotopes (particularly $^3$H) and

atmospheric gases such as the chlorofluorocarbons (CFCs), $SF_6$, and $^{85}$Kr, whose atmospheric

concentrations have increased over recent decades (e.g. Cook and Bohlke, 2000;

Morgenstern et al., 2010; Kirchner et al., 2010; Yang et al., 2011). Estimation of MTTs is

commonly determined via Lumped Parameter Models (LPMs) that describe the distribution

of water with different ages or tracer concentrations in simplified aquifer geometries. With

LPMs, the MTT at the time of sampling is evaluated by comparing the input history of a

tracer in precipitation to the measured concentration of that tracer within a stream via the

convolution integral (Maloszewski and Zuber, 1982, 1996; Maloszewski et al., 1983).

Determining MTTs from stable isotopes or major ion concentrations relies on tracking the

delay and dampening of their seasonal variations between precipitation and discharge.

However, use of these tracers typically requires sub-weekly sampling over time periods





equal to or exceeding that of the transit times (Timbe et al., 2015).  In addition, these
tracers become ineffective when transit times exceed 4 to 5 years as the initial variations in
rainfall are progressively dampened to below detection limits (Stewart et al., 2010).

Gaseous tracers (e.g. $^3$He, chlorofluorocarbons, SF$_6$) are effective in determining residence
times of groundwater that is separated from the atmosphere (Cook and Bohlke, 2000) but
are difficult to apply to surface water due to gas exchange.  With a half-life of 12.32 years,
tritium ($^3$H) has been used to estimate MTTs of up to about 150 years (e.g. Morgenstern et
al., 2010).  Unlike other radioactive tracers (such as $^{14}$C), $^3$H is part of the water molecule
and its activities are affected only by radioactive decay and dispersion and not by water-
rock interaction. Also, because $^3$H activities are not affected by processes in the unsaturated
zone, MTTs estimated using $^3$H reflect both recharge through the unsaturated zone and flow
in the groundwater system.

Utilisation of $^3$H as a tracer has been facilitated by the fact that $^3$H activities of rainfall have
been measured globally for several decades (IAEA, 2016), including in southeast Australia
(Tadros et al., 2014).  Due to atmospheric nuclear testing, $^3$H activities in rainfall peaked
during the 1950s and 1960s (the "bomb-pulse"), particularly in the northern Hemisphere
(Tadros et al., 2014).  As a result, single $^3$H activities of waters in the Northern Hemisphere
yield non-unique MTTs, although MTTs may still be estimated using time series $^3$H data.  In
the Southern Hemisphere, bomb-pulse $^3$H activities have declined to levels below that of
modern rainfall due to removal by precipitation and radioactive decay (Morgenstern et al.,
2010).  As a consequence, transit times can, in most cases, now be determined from single
$^3$H measurements (Morgenstern et al., 2010; Morgenstern and Daughney, 2012) in an
analogous manner to how other isotopic tracers (e.g., $^{14}$C or $^{36}$Cl) are used in regional
groundwater systems.

Use of LPMs to evaluate MTTs carries a number of uncertainties, including deciding on
which LPM to employ, aggregation error, the tracer input history, and analytical error.  In
the past, due to remnant bomb-pulse $^3$H activities, the choice of LPM had a very large
impact on the calculated MTTs. However, the gradual reduction of the bomb-pulse $^3$H over





time allowed the appropriateness of the LPM to be evaluated by time-series [3]H
measurements (e.g., Maloszewski and Zuber, 1982; Zuber et al., 2005). Due to the
attenuation of the [3]H bomb-pulse in the southern hemisphere, the calculated MTTs are now

less sensitive to the choice of LPM employed. However, this also results in LPMs no longer
being able to be evaluated by time-series [3]H measurements (Cartwright and Morgenstern,
2016a). As a consequence, LPMs must typically be assigned based upon knowledge of the
geometry of the flow system and/or information from previous time-series studies.

Rivers can receive water from numerous stores, including groundwater, tributaries, soil

water, and perched aquifers, each of which may have different MTTs. MTTs estimated using
geochemical tracers in the aggregated water tends to underestimate the actual MTT (i.e.
that which would be calculated using the weighted average of each store). This is known as
the aggregation error (Kirchner, 2016a, b; Stewart et al., 2016) and it increases as the
difference between the transit times of the individual end-members also increases.

However, for transit times estimated from single [3]H activities, the aggregation error
decreases with an increasing number of end-members (Cartwright and Morgenstern,
2016b).

### 1.2. Controls on Mean Transit Times

A relatively large volume of work has been conducted to understand the catchment

attributes that control MTTs. Being able to identify such controls is important as it would
allow first order estimates of MTTs to be made in similar catchments for which detailed
geochemical tracer data do not exist. Previous studies have identified catchment size (e.g.
McGlynn et al., 2003; Hrachowitz et al., 2010), groundwater storage volumes (e.g. Ma and
Yamanaka, 2016), topography (e.g. McGuire et al., 2005), bedrock permeability (e.g. Hale

and McDonnell, 2016), drainage density (e.g. Hrachowitz et al., 2009), forest cover (e.g.
Tetzlaff et al., 2007), and soils (e.g. Tetzlaff et al., 2009) as important controls. However, no
single attribute has been shown to be the dominant control at all locations. In other
catchments, correlations between [3]H activities and major ion geochemistry or the runoff





coefficient (the proportion of rainfall exported from the catchment by the stream) allow

first order estimates of MTTs to be made (Morgenstern et al., 2010; Cartwright and

Morgenstern, 2015, 2016a).

## 2. Objectives

This study focuses on six headwater catchments in the Otway Ranges of southeast Australia.
Largely contained with the Great Otway National Park, the Otway Ranges hold ecological,

cultural, historical and recreational significance.  In addition, these headwater streams

contribute a significant portion of flow to the Gellibrand River, which acts as a water source

for several towns, supports important aquatic and terrestrial fauna, and provides water for

agricultural.  Despite their significance, the headwater catchments of the Otway Ranges face

a number of threats, including urbanisation, clearing of native vegetation, drought and

bushfire, all of which have the potential to impact the quantity and quality of water within

the streams.

The primary objective of this study is to determine the MTTs in these headwater streams to

enable estimates of groundwater stores, lag times, controls on stream flow generation, and

impact of land use on stream water quality.  If the streams are to be protected, being able

to answer this question is of utmost importance.  Secondary objectives include: 1) assessing

uncertainties in the MTTs, 2) evaluating potential water inputs into the streams, 3) assessing

potential controls on MTTs, 3) investigating possible proxies for $^3$H, and 4) appraising water

quality impacts within the catchments.  It is expected that the results of this investigation will

facilitate greater understanding of headwater streams not only within the Otway Ranges but in

similar catchments worldwide.

## 3. Study Area

The Otway Ranges are located in south-central Victoria, Australia, approximately 150 km

southwest of Melbourne (Fig. 1).  The region has a temperate climate, with average annual

rainfall varying from approximately 1,000 mm at Gellibrand and Forrest to approximately

1,600 mm at Mount Sabine (Department of Environment, Land, Water and Planning





(DELWP), 2017) (Fig. 1).  The majority of rainfall occurs during the austral winter months
(July to September) and, during summer months, potential evaporation exceeds
precipitation (Bureau of Meteorology, 2016).  The Otway Ranges are dominated by
eucalyptus forest but include some production forestry.

The Gellibrand River is one of the larger river systems draining the region.  It flows west-
southwest for approximately 100 km from its highest point in the Otway Ranges before
discharging into the Southern Ocean near Princetown.  This study focuses on six headwater
sub-catchments of the Gellibrand River: Lardners Creek, Love Creek, Porcupine Creek, Ten
Mile Creek, Yahoo Creek and the Gellibrand River upstream of James Access (Fig. 1).

Porcupine Creek, Ten Mile Creek and Yahoo Creek are the main tributaries to Love Creek
which, together with Lardners Creek, discharge into the Gellibrand River near Gellibrand
(Fig. 1).

The geology of the study area has been discussed extensively by Tickell et al. (1991).  The
basement comprises the early-Cretaceous Otway Group, which consists primarily of

volcanogenic sandstone and mudstone with minor amounts of shale, siltstone, and coal.
The Otway Group is considered to be a poor aquifer and crops out across most of the
Lardners Creek and Gellibrand River Catchments, as well as within the higher elevation
areas of the Yahoo Creek and Ten Mile Creek catchments (Fig. 1).

The Otway Group is uncomformably overlain by a sequence of Tertiary sediments

comprising the Eastern View Formation, the Demons Bluff Formation, the Clifton Formation
and the Gellibrand Marl.  The Eastern View Formation is composed of three sand and gravel
units that collectively form the Lower Tertiary Aquifer.  These sediments crop out at various
locations across the study area including at the Barongarook High (Fig. 1), which is the
primary recharge area for the aquifer (Stanley, 1991; Petrides and Cartwright, 2006).

The Eastern View Formation is overlain by the Demons Bluff Formation, which is a
calcareous silt having negligible permeability.  The formation crops out sparsely within the
study area, mainly along Yahoo and Ten Mile Creeks.  Overlying this unit is the Clifton





Formation, which forms a minor aquifer and is comprised primarily of limonitic sand and

gravel.  This unit crops out along Porcupine, Ten Mile, Yahoo and Love Creeks.  The Clifton

Formation is overlain by the Gellibrand Marl, which consists of approximately 200 to 300 m

of calcareous silt.  The marl crops out extensively within the Love Creek and Porcupine

Creek catchments and acts as a regional aquitard.  Along Love Creek and parts of the

Gellibrand River, the Tertiary units have been intruded by the Yaugher Volcanics, which

consist primarily of basalt, tuff and volcanic breccia.  Deposits of alluvium are present along

most of the stream courses, particularly Porcupine Creek and Love Creek.

Regional groundwater flows from the recharge area in the Barongarook High to the south

and southwest (Leonard et al., 1981; Stanley, 1991; SKM; 2012; Atkinson et al., 2014).

Additional, localised recharge may occur elsewhere across the study area, particularly in

those areas where the Eastern View Formation crops out.  Regional groundwater discharges

into the Gellibrand River, Love Creek, Porcupine Creek, Ten Mile Creek and Yahoo Creek

(Hebblethwaite and James, 1990; SKM, 2012; Atkinson et al., 2013; Costelloe et al., 2015).

In the higher elevations of the study area, including the upper reaches of Lardners Creek,

the regional water table is likely to be below the base of the streambed (Costelloe et al.,

2015).  Based upon $^{14}$C and $^{3}$H activities, residence times of the regional groundwater are

between 100 and 10,000 years (Petrides and Cartwright, 2012; Atkinson et al., 2014).

## 4. Methodology

### 4.1. Water Sampling

River water samples were collected from eight locations in the catchments (Fig. 1).  Two

locations were sampled in the Lardners Creek Catchment: at an active gauging station on

Lardners Creek (Lardners Gauge) maintained by DELWP (Site ID 235210) and from the

Lardners Creek East Branch (Upper Lardners), located approximately 3.5 km upstream from

Lardners Gauge.  Love Creek was sampled at two locations: at Kawarren (Love Creek

Kawarren), located approximately 1 km upstream of a DELWP gauging station (Site ID

235234) and at the Wonga Road crossing (Love Creek Wonga), which is located





approximately 4.5 km downstream of Kawarren.  River water samples were collected from

the Gellibrand River, Porcupine Creek, Ten Mile Creek and Yahoo Creek at the sites of

former DELWP gauging stations (Site IDs 235235, 235241, 235239 and 235240, respectively).

River water samples were collected from each site in July 2014, September 2014, March

2015 and September 2015.  An additional round of river water samples was collected from

Lardners Gauge, Porcupine Creek, Ten Mile Creek and Love Creek Kawarren in November

2015.  The water samples were collected from close to the centre of the streams using a

polyethylene container fixed to an extendable pole.  Additional data for the Gellibrand River

at James Access is from Atkinson (2014).

A single precipitation sample was collected from Birnam in the Otway Ranges near Ten Mile

Creek (Fig. 1) in September 2014 using a rainfall collector.  The collector consisted of a

polyethylene storage container equipped with a funnel positioned approximately 0.5 m

above ground level.  Prior to collection of the precipitation sample, the collector had been in

the field for 78 days, during which time approximately 198 mm of rainfall was recorded at

Forrest while 431 mm of rainfall was recorded at Mount Sabine (DELWP, 2017).

**4.2. Discharge Determination**

Discharge at the time of sampling was determined for each of the eight locations with the

exception of the Upper Lardners, which is ungauged.  Discharge is monitored by DELWP at

gauging stations located on Lardners Creek (Site ID 235210) and at Love Creek (Site ID

235234).  At the Gellibrand River sampling site (James Access), discharge was estimated

using a correlation ($R^2$ = 0.97) between discharge at the former gauging station at this

location and that at the existing Upper Gellibrand River gauging station (Site ID 235202),

located approximately 7 km upstream (Fig. 1).  Likewise, discharge at the Porcupine Creek,

Ten Mile Creek and Yahoo Creek sampling sites was estimated using correlations ($R^2$ = 0.95,

$R^2$ = 0.77 and $R^2$ = 0.84, respectively)  between discharge at the former gauging stations at

these locations and that at the Love Creek gauging station.



**Analytical Techniques**

The EC of the river water and precipitation samples was measured in the field using a calibrated TPS® hand-held water quality meter and probe. The EC measurements have a precision of 1 µS/cm. The river water and precipitation samples were analysed for cations,

anions and $^3$H (Supplement). Cation concentrations were measured at Monash University using a ThermoFinnigan ICP-OES on samples that had first been filtered through 0.45 µm cellulose nitrate filters and acidified to a pH < 2 using double-distilled 16 M $HNO_3$. Anion concentrations were measured at Monash University on filtered, un-acidified samples using a Metrohm ion chromatograph (IC). The precision of the cation and anion analyses, based

upon replicate sample analysis, is ± 2% while the accuracy, based on analysis of certified water standards, is ± 5%. Duplicate samples were prepared and analysed at a rate of approximately one per sampling event. Total dissolved solids (TDS) concentrations were determined by summing the concentrations of cations and anions.

$^3$H analysis was conducted at the GNS Water Dating Laboratory in Lower Hutt, New Zealand.

The samples were distilled and electrolytically enriched prior to analysis by liquid scintillation counting, as described by Morgenstern and Taylor (2009). $^3$H activities are expressed in tritium units (TU) with a relative uncertainty of ± 2% and a quantification limit of 0.02 TU. Correlations between geochemical variables are discussed below. A reasonably strong correlation is viewed to exist if the correlation coefficient ($R^2$) is greater than 0.7.

**4.3. Calculating Mean Transit Times**

Groundwater takes a myriad of flow paths between the recharge areas to where it discharges. Consequently, groundwater does not have a discrete transit time but instead has a distribution of transit times. The MTT may be estimated using LPMs. A number of commonly-used LPMs have been developed (e.g. Maloszewski and Zuber, 1982, 1992; Cook

and Bohlke, 2000; Maloszewski, 2000; Zuber et al., 2005). In each of these models, the concentration of a tracer (e.g. $^3$H) sampled from a stream or bore at time $t$ ($C_0(t)$) is related to the input ($C_i$) of that tracer at the recharge area via the convolution integral:



$$C_0(t) = \int_0^\infty C_i\,(t - \mathrm{T})\,g\,(\mathrm{T})e^{-\lambda\mathrm{T}}d\mathrm{T} \qquad\qquad (1)$$

where T is the transit time, t – T is the time that the groundwater entered the flow system,

λ is the decay constant (0.0563 yr[-1] for [3]H) and g (T) is the exit age distribution function, for

which closed form analytical solutions have been derived (e.g. Maloszewski and Zuber,

1982; Maloszewski and Zuber, 1996; Kinzelbach et al., 2002).

As discussed earlier, the use of single [3]H activities to estimate MTTs requires that an LPM be

assigned.  In this investigation, two LPMs were utilised: the Exponential Piston-Flow Mode

(EPM) and the Dispersion Model (DM).  These are among the most commonly utilised LPMs

(McGuire and McDonnell, 2006) and are discussed briefly below.

The EPM describes aquifers with two segments of flow: a portion with an exponential age

distribution, and a piston-flow portion.  Conceptually, this model most closely applies to

aquifers that are unconfined in the recharge area (the exponential segment) and confined

(the piston flow segment) at lower elevations, where there is little to no recharge.  The

Yahoo Creek, Ten Mile Creek, Love Creek and Porcupine Creek Catchments can potentially

be described by this model, as recharge to the Lower Tertiary Aquifer occurs in the higher

elevations of the catchments, but is limited in lower elevation areas by the presence of the

Gellibrand Marl and/or the Demons Bluff Formation.  The EPM has also been applied to

unconfined aquifers, as recharge through the unsaturated zone resembles piston flow while

flow within the aquifer resembles exponential flow (e.g. Cook and Bohlke, 2000;

Morgenstern et al., 2010; Cartwright and Morgenstern, 2015; Cartwright and Morgenstern,

2016a).  Utilisation of the EPM requires defining a value for the EPM ratio, which represents

the relative contribution of the exponential and piston flow model components (Jurgens et

al., 2012).  The EPM ratio is defined as $1/f$ - 1, where $f$ is the proportion of aquifer volume

exhibiting exponential flow.

The Dispersion Model (DM) is based on the one-dimensional advection-dispersion equation

for a semi-infinite medium (Jurgens et al., 2012).  While the DM can be applied to a wide

variety of aquifer configurations, conceptually it is probably less realistic than other LPMs.





Nonetheless, it has been successfully used to predict tracer concentrations over time in a number of flow systems (e.g. Maloszewski, 2000). Utilisation of this model requires defining the value of the dispersion parameter, $D_p$ (the ratio of dispersion to advection), which is seldom known *a priori*.

MTTs were estimated using TracerLPM (Jurgens et al., 2012) and a $^3$H record for rainfall modified from the Melbourne rainfall record. Modern rainfall in Melbourne (located approximately 150 km from the study area) has a $^3$H activity of approximately 3.0 TU, while modern rainfall in the study area has an expected $^3$H activity of approximately 2.8 TU (Tadros et al., 2014). Thus, a $^3$H value of 2.8 TU was utilised for modern (2010 to 2016)

rainfall, as well as for the years prior to the atmospheric nuclear tests (pre-1951). The $^3$H activities for rainfall between 1950 and 2009 were decreased by 6.7% to account for the expected difference in $^3$H activities within the Otways Ranges relative to Melbourne. MTTs were estimated by matching the predicted $^3$H activities from the LPMs to the observed $^3$H activities of the samples.

**4.4. Determining Catchment Attributes**

Catchment attributes were determined using ArcGIS 10.2 (ESRI, 2013) in combination with ground surface elevation contours, bedrock geology, stream courses, and land use data (DataSearch Victoria, 2015). A 20 m digital elevation model (DEM) of the study area was constructed, from which catchment area, drainage density, and average topographic slope

for each catchment were determined. In addition, runoff coefficients were calculated using discharge data for each of the catchments (except Upper Lardners) for the period of March 1986 to July 1990, the only interval for which contiguous discharge data are available for each catchment. In the runoff coefficient calculations, an average annual rainfall of 1.3 m was assumed for each catchment.



## 5. Results

### 5.1. River Discharge

Figure 2 illustrates the discharge conditions under which sampling occurred relative to the flow duration curves for each catchment except for Upper Lardners.  Samples were generally collected between the 10th and 100th percentiles of discharge.  Figure 3 shows discharge at Lardners Gauge and Love Creek over the sampling period.  Samples were collected during recession periods after high discharge events or during base flow conditions.  Overland flow was not observed during any of the sampling events.

Discharge was highest during July 2014 (Supplement), ranging from 8.6 x 10$^3$ m$^3$ day$^{-1}$ at Ten Mile Creek to 255.2 x 10$^3$ m$^3$ day$^{-1}$ in the Gellibrand River at James Access.  Discharge was lowest during March and November 2015, ranging from 0.1 x 10$^3$ m$^3$ day$^{-1}$ at Ten Mile Creek to 8.8 x 10$^3$ m$^3$ day$^{-1}$ at James Access.

### 5.2. Tritium Activities

The precipitation sample collected from near Ten Mile Creek in September 2014 had a tritium activity of 2.45 TU, which is near the low end of the predicted range (2.4 to 3.2 TU) of $^3$H activities of modern rainfall for this area (Tadros et al., 2014).  This $^3$H activity is also below the values of 2.70 and 2.76 TU from 9 to 12 month samples of rainfall in the Melbourne area (Atkinson, 2014; Cartwright, unpublished data), and 2.85 to 2.99 TU for 9 to 17 month samples for rainfall in the Ovens River Catchment in northern Victoria (Cartwright and Morgenstern, 2015). The lower than expected $^3$H activity from the Otway sample is probably due to the sample representing rainfall of only part of the year.

Tritium activities in the river water samples are all lower than those of modern rainfall and ranged from 0.20 TU at Porcupine Creek in March 2015 to 2.14 TU at Yahoo Creek in July 2014 (Supplement).  In general, $^3$H activities were highest at high stream flows (July 2014) and lowest at low stream flows (March and November 2015).  The $^3$H activities of Love Creek were relatively similar between the upstream and downstream sampling locations during each sampling event.  At Lardners Creek, $^3$H activities decreased downstream during



the two highest discharges (July 2014 and September 2015) but increased downstream during lower discharges (March and November 2015).

The range of $^3$H activities was most variable at Porcupine Creek (0.20 to 1.97 TU), followed

by Yahoo Creek (0.43 to 2.14 TU), Love Creek Kawarren (0.48 to 1.91 TU), Love Creek Wonga (0.55 to 1.88 TU), Ten Mile Creek (0.44 to 1.74 TU), Upper Lardners (1.54 to 1.99 TU), the Gellibrand River at James Access (1.73 to 2.08 TU) and Lardners Gauge (1.64 to 1.97 TU) (Fig. 4). Thus, while the highest $^3$H activity values were similar across all catchments, the lower values varied considerably.

There is a reasonably good correlation ($R^2$ = 0.75) between $^3$H activities and discharge (Q) for the catchments as a whole (Fig. 4), whereby $^3$H = 0.2613 ln (Q) + 0.8973. The $^3$H activities increase with increasing discharge (Fig. 4) up to a threshold of approximately $10^4$ $m^3\,day^{-1}$, above which $^3$H activities do not increase appreciably above ~2.0 TU. The maximum $^3$H activity (2.14 TU) in the rivers is less than both the predicted and measured $^3$H

activities of rainfall in southeast Australia. However, it is within the range of $^3$H activities of 1.80 to 2.25 TU for soil pipe water in higher elevation areas of the Gellibrand River Catchment (Atkinson, 2014).

### 5.3. Major Ion Geochemistry

River water geochemistry is similar across all catchments and is dominated by Na, Cl and

HCO$_3$. TDS concentrations are generally less than 100 mg/L at Lardners Gauge, Upper Lardners and the Gellibrand River at James Access but typically exceed 200 mg/L in Love Creek, Porcupine Creek, Ten Mile Creek and Yahoo Creek. TDS concentrations generally increase downstream at Lardners and Love Creeks and are inversely correlated with discharge in all catchments.

At Love Creek, Ten Mile Creek, Yahoo Creek and Upper Lardners, there is no correlation between $^3$H activities and EC, TDS or major ion concentrations (Fig. 5). However, at Porcupine Creek, there is a strong correlation ($R^2 > 0.95$) between $^3$H activities and EC, TDS, and all major ion concentrations with the exception of chloride, nitrate and sulphate. In




addition, there is a relatively strong correlation ($R^2$ = 0.84) between $^3$H activities and TDS at

Lardners Gauge (Fig. 5).

At Upper Lardners, the Gellibrand River at James Access and Ten Mile Creek, there is a

strong correlation ($R^2$ > 0.90) between nitrate concentration and $^3$H activities (Fig. 6a).  The

range of nitrate concentrations (0.08 to 2.0 mg/L) were relatively similar during each

sampling event across all catchments except for in July 2014, when nitrate concentrations

exceeded 3 mg/L at Love Creek Kawarren and Love Creek Wonga.  A similar correlation

exists between sulphate concentrations and $^3$H activities at the Gellibrand River at James

Access and at Upper Lardners, but not at Ten Mile Creek (Fig 6b).  However, sulphate

concentrations at these locations are lower than they are in the other catchments.

### 5.4. Catchment Attributes

Love Creek Wonga has the largest drainage area of the six catchments at approximately 91.7

km$^2$ (Table 1).  This drainage area includes the Love Creek Kawarren, Yahoo Creek, Ten Mile

Creek and Porcupine Creek sub-catchments, which have drainage areas of 74.4 km$^2$, 16.6

km$^2$, 9.6 km$^2$ and 33.6 km$^2$, respectively.  Lardners Gauge has a drainage area of 51.6 km$^2$,

which includes the Lardner Creek East Branch (Upper Lardners) sub-catchment with an area

of approximately 20 km$^2$.  The Gellibrand River Catchment upstream of James Access has

the second largest drainage area of approximately 81.0 km$^2$.

Drainage densities within the six catchments are relatively similar and range from

approximately 8.7 x 10$^{-4}$ m m$^{-2}$ at Yahoo Creek to 1 x 10$^{-3}$ m m$^{-2}$ at Lardners Gauge and

Upper Lardners.  Forest cover is lowest in the Love Creek Wonga and Love Creek Kawarren

catchments, at approximately 78% and 82%, respectively.  Within the remaining

catchments, forest cover varies from 88% within the Porcupine Creek and Ten Mile Creek

catchments, 91 to 92% in in the Lardners Creek catchments, and 95% in the Gellibrand River

and Yahoo Creek catchments.  Average slope is approximately 11° in the Lardners Gauge,

Upper Lardners and Gellibrand River at James Access Catchments and 8.6° in the Yahoo





Creek Catchment.  Within the Ten Mile Creek, Porcupine Creek, Love Creek Kawarren and

Love Creek Wonga catchments, average slope varies from 5.7 to 6.7°.

Based upon an average annual rainfall of approximately 1.3 m across all catchments, runoff

coefficients range from 33% and 39% at Lardners Creek and the Gellibrand River at James

Access, respectively, to 9% to 12% at Porcupine Creek, Ten Mile Creek, Yahoo Creek and

Love Creek.

There are either weak or no correlations ($R^2 \leq 0.6$) between $^3$H activities and catchment

area, drainage density or forest cover (Table 2).  However, there are strong positive

correlations between $^3$H activities and the runoff coefficient ($R^2 = 0.94$) (Fig. 7) and between

$^3$H activities and average topographic slope ($R^2 = 0.87$), but only for samples collected during

March 2015, when stream flow was generally lowest.  However, these correlations are

based upon only a small number of samples.  Further, the results may be skewed by the

data for Lardners Gauge and the Gellibrand River at James Access catchments, which have

much higher runoff coefficients and slopes than the other catchments.

## 6.  Discussion

The discharge, tritium and major ion geochemistry data, in combination with catchment

attributes, allow an assessment of MTTs, uncertainties in the MTTs, groundwater recharge

and water quality impacts.

### 6.1. Sources of Baseflow

Each of the river water samples was collected during baseflow conditions or during

recession periods after high discharge events.  Furthermore, there are few systematic

variations in major ion geochemistry with stream discharge that would suggest that there is

significant dilution of groundwater inflows with recent rainfall during the sampling periods.

The flow system may therefore be viewed as a continuum that is dominated by older

groundwater inflows at low flows and progressively shallower and younger stores of water

(such as soil water or perched groundwater) that are mobilised during wetter periods.  If





this is the case, the system may be modelled using a single LPM.  If there were some dilution by recent rainfall, this approach yields the minimum MTT of the baseflow component.

### 6.2. Mean Transit Times

MTTs in the headwaters catchments were estimated using the EPM and the DM.  Initially, an

EPM ratio of 0.33 (75 % exponential flow) was utilised, as this value has been shown to be effective in modelling $^3$H time series in catchments of New Zealand (Morgenstern and Daughney, 2012, Morgenstern et al. 2010).  To assess the effects of adopting different LPMs, MTTs were also determined using the EPM with EPM ratios of 1.0 (50 % exponential flow) and 3.0 (25 % exponential flow) and the DM with Dp of 0.05 and 0.5. This range of Dp values

applies to most flow systems of this scale (Zuber and Maloszewski, 2001; Gelhar et al., 1992).

MTTs ranged from approximately 7 years at Yahoo Creek in July 2015 to 234 years at Porcupine Creek in March 2015 (Table 3).  In general, the lowest estimates of MTTs were derived using the EPM with an EPM ratio = 3.0 while the highest estimates of MTTs were

derived using the DM with $D_p$ = 0.5.   MTTs estimated with all models were relatively similar for $^3$H activities greater than ~1.00 TU (Fig. 8).  However, as $^3$H activities decrease below this value, the relative difference between the estimates increases.  At the lowest reported $^3$H activity of 0.20 TU, the relative difference across the range of transit times is approximately 164 years (110%).

At Lardners Gauge, the Gellibrand River at James Access, Porcupine Creek and Love Creek, the samples collected at the highest flow rates have MTTs that are slightly higher (older) than that of the samples collected at the second highest discharge (Fig. 9).  Whether this reflects changes to the flow system or is due to uncertainties in the MTTs (discussed below) is not certain.

In the individual catchments, MTTs for Lardners Gauge, Upper Lardners and the Gellibrand River at James Access were relatively similar and ranged from approximately 7 to 26 years. In contrast, MTTs for Porcupine Creek ranged from approximately 7 to 234 years, while

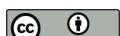



those for Ten Mile Creek, Yahoo Creek and Love Creek ranged from approximately 13 to 149

years, 7 to 154 years and 10 to 141 years, respectively.  In all catchments, the highest

(oldest) MTTs are associated with the lowest discharge conditions (March 2015) while the

lowest (youngest) MTTs are associated with higher discharge conditions (July 2014 and

September 2015) (Fig. 9).  The low discharge MTTs at Porcupine Creek, Ten Mile Creek,

Yahoo Creek and Love Creek are considerably greater than the average MTT of 15 ± 22 years

for headwater catchments worldwide reported by Stewart et al. (2010).

The MTTs for a given water sample, particularly where $^3$H activities are less than ~0.5 TU

(Fig. 8) vary considerably.  However, as discussed earlier, it is not possible to assess the most

suitable LPM.  The EPM with an EPM ratio of 3.0 and the DM with a $D_p$ value of 0.05

simulate groundwater having a large component of piston flow and, for this reason, are

most likely less realistic representations of the flow systems.  In contrast, MTTs derived

using the EPM with an EPM ratio of 0.33 and the DM with a $D_p$ value of 0.5 are relatively

similar across the full range of $^3$H activities.  The EPM with an EPM ratio of 1.0 produces

transit time estimates that fall approximately midway between the other four models.

Because of the remnant bomb pulse $^3$H, a few samples with $^3$H activities between 1.2 to 1.7

TU yield MTTs that are non-unique for models with high piston flow components (i.e., the

EPM with EPM ratio = 3.0 and the DM with $D_p$ = 0.05; Table 3, Fig. 8).

### 6.3. Uncertainties in the MTT Estimates

A number of uncertainties exist within the MTT estimates: a) potential aggregation error, b)

uncertainty in the $^3$H activity of rainfall, and c) analytical uncertainty in the laboratory-

derived $^3$H activities.  Each of these uncertainties are discussed below.

**6.3.1.    Aggregation Error**

Aggregation of water with different MTTs introduces uncertainty in the calculation of MTTs

(Kirchner, 2016a, b; Stewart et al., 2016).  In general, MTTs calculated from the aggregated

water underestimate the MTT that would be calculated from the weighted average of the

end-members. Quantifying this potential error is not straightforward, however, as the





number of inputs (including tributaries) contributing to total stream flow at a given sampling

location is generally unknown, as are the transit times of these inputs.  Stewart et al. (2016)

indicate that aggregation error becomes significant when MTTs determined using $^3$H and

simple LPMs exceed approximately 6 to 12 years.  As most of the MTTs derived in this study

are several decades (or longer), it is possible that the calculated MTTs underestimate the

true MTTs.

To evaluate this potential error, true MTTs were estimated for Love Creek Kawarren using

the discharge data, $^3$H activities, and MTTs for Porcupine, Ten Mile and Yahoo Creeks,

whose confluence is located a short distance upstream of the Love Creek Kawarren sampling

point.  These were then compared with the MTT calculated from the measured $^3$H activities

at that site. The analysis used the EPM with an EPM ratio of 1.0 (Table 3), but similar results

were obtained with the other LPMs.  Inputs from these three streams contribute 77 to 82%

of total stream flow at Love Creek Kawarren.  The remaining portion of flow is contributed

by undefined inputs that may include both groundwater inflow and smaller tributaries.  True

MTTs at Love Creek Kawarren were calculated using the relationship (modified after Stewart

et al. (2016)):

$$MTT_{LK} (true) = a * MTT_{PC} + b * MTT_{TC} + c * MTT_{YC} + MTT_{UI} \hspace{2cm} (2)$$

Where a, b, c and d represent the fraction of total flow contributed by Porcupine Creek (PC),

Ten Mile Creek (TC), Yahoo Creek (YC) and the undefined inputs (UI), and $MTT_{PC}$, $MTT_{TC}$,

$MTT_{YC}$ and $MTT_{UI}$ are the MTTs for these inputs.  $MTT_{UI}$ was determined from the calculated

$^3$H activity of the undefined inputs, which was estimated through $^3$H mass balance and the

same LPM.

During March 2015, the sample MTT at Love Creek Kawarren over-estimated the true MTT

by approximately by approximately 3.7 years or 4% (Table 4).  At all other times, sample

MTTs underestimated true MTTs by approximately 3.9 to 7.4 years (18 to 37%).  If the

system aggregated more stores of water with a similar range of $^3$H activities, the

aggregation error is likely to be less (Cartwright and Morgenstern, 2016a).  While the



aggregation error introduces uncertainties, it does not alter the conclusion that the MTTs
are years to decades.

### 6.3.2.    $^3$H activity of Rainfall

There is obviously some uncertainty in the rainfall $^3$H activities and Tadros et al. (2014)
proposed that modern rainfall $^3$H activities were 2.4 to 2.8 TU to the west of the study area
and 2.8 to 3.2 TU to the east.  The single rainfall sample from near Ten Mile Creek in
September 2014 had a $^3$H activity of 2.45 TU, which is near the low end of the range.
However, this sample was collected over a period of only 78 days and may therefore not be

representative of annual rainfall.  To assess the effect of uncertainties in rainfall $^3$H
activities, MTTs were recalculated assuming that modern and pre-1950 rainfall had a $^3$H
activity of either 2.4 TU or 3.2 TU with the $^3$H activities of the intervening years adjusted
proportionally.  Again, this used the EPM with an EPM ratio of 1.0 but the effect is similar in
the other models.

The relative difference between MTTs calculated from the three rainfall records is generally
highest (up to 140%) when $^3$H activities are greater than ~1 TU but decreases with
decreasing $^3$H activities (Fig. 10).  However, the high relative differences in MTTs at $^3$H
activities greater than 1 TU is, in part, offset by low absolute differences.  For $^3$H activities
less than ~0.6 TU, the variation in the rainfall input results in less than 4% difference in

MTTs.  These results indicate that uncertainties in the rainfall $^3$H activities are relatively
unimportant for waters with very low or very high $^3$H activities.

The catchments are located in a relatively small geographic area and, for this reason, likely
receive rainfall from the same weather systems. Thus, $^3$H inputs are likely to be the closely
similar in each catchment.  If this is the case, uncertainties in the rainfall $^3$H activities may

result in uncertainties in the absolute MTT estimates but will have less impact on the
relative differences in MTTs at different times in the same catchment, or between
catchments.





### 6.3.3. Analytical Uncertainty

The $^3$H activities have a laboratory analytical uncertainty ranging from ± 0.02 to 0.04 TU.

The ± 0.04 TU uncertainty for the sample with the highest $^3$H activity (2.14 TU) results in a

maximum uncertainty in the MTT of ± 0.9 years, depending on the LPM utilised.  Likewise,

the ± 0.02 TU uncertainty for the sample having the lowest $^3$H activity (0.20 TU) results in a

maximum uncertainty in the MTT of ± 10 years. Relative to aggregation error and

uncertainty in the rainfall record, analytical uncertainty is relatively minor in significance.

In summary, the MTTs presented in Table 3 are subject to several uncertainties, including

uncertainties about the most appropriate LPM to use, the aggregation error, uncertainty in

rainfall $^3$H inputs, and analytical error.  Uncertainties in the LPM and the aggregation error

are probably most significant, especially at intermediate flow rates, when $^3$H activities

within the streams are most variable.

### 6.4. Variability in MTTs at Porcupine Creek

Between January 1990 and January 1994, DELWP measured EC and discharge on a monthly

basis at the former gauging station (Site ID 235241) on Porcupine Creek.  These data, in

combination with a strong correlation ($R^2$ = 0.96) between MTTs and EC at this location,

given by MTT = 1.362$e^{0.0061*EC}$ allow a first order estimation of MTTs within the stream over

this four year period (Fig. 11). The estimated MTTs range from approximately 3 to 50 years

and exhibit a seasonal pattern whereby the highest MTTs generally correspond to low,

summer flows and the lowest MTTs correspond to high, winter flows.  Although based upon

a limited number of samples, these results demonstrate the high variability of transit times

within the catchment and the value of finding proxy analytes for $^3$H.

### 6.5. Groundwater Recharge at the Barongarook High

The volume of groundwater (V) stored within an aquifer can be estimated from the

relationship:

$$V = Q_R * MTT_R \qquad\qquad\qquad (3)$$





where $Q_R$ represents river discharge and $MTT_R$ is the MTT of the river water (Morgenstern et al., 2010). The relationship between $MTT_R$ and $Q_R$ at Ten Mile and Yahoo Creeks is defined by the best fit correlation between the two parameters (Fig. 9):

$$MTT = 86.77 * e^{-2E-04\,Q} \quad (R^2 = 0.99, \text{Ten Mile Creek}) \qquad (4)$$

$$MTT = 4847 * Q^{-0.64} \quad (R^2 = 0.98, \text{Yahoo Creek}) \qquad (5)$$

Using the above relationships and river discharge at the time of sampling, the volume of groundwater stored within the Ten Mile Catchment was approximately 5,500 m$^3$ in March 2015 and 42,000 m$^3$ in July 2014. Likewise, at Yahoo Creek, groundwater volumes varied from approximately 15,300 m$^3$ in March 2015 to 65,800 m$^3$ in July 2014. If it assumed that the difference between these values represents the average volume of water recharged to the aquifer in a year, then groundwater recharge can be estimated from average annual rainfall (approximately 1.3 m year$^{-1}$) and the size of the recharge area. If groundwater within the two catchments is recharged entirely through the Eastern View Formation, which has outcrop areas of approximately 3,467,400 m$^2$ and 2,588,900 m$^2$ respectively, groundwater recharge is approximately 0.8 % (11 mm year$^{-1}$) in the Ten Mile Creek and 1.5 % (20 mm year$^{-1}$) in the Yahoo Creek catchments

The above calculations were based on the MTTs from the EPM with an EPM ratio of 1.0. If an EPM ratio of 3.0 is utilised, the same recharge rates are obtained. Using the DM and a $D_p$ value of 0.5 leads to recharge estimates of 1.3 % and 1.4 %. These recharge estimates are considerably less than those estimated by Leonard et al. (1981) at 17 %, Witebsky et al. (1992) at 8 %, and Teng (1996) at 9 %. However, they are comparable to those derived for other parts of southeast Australia (e.g. Cook et al., 1994; Cartwright et al., 2007). This exercise demonstrates the potential for using MTTs to estimate groundwater recharge.

### 6.6. Impacts to River Water Quality

Nitrate concentrations increase with a corresponding increase in $^3$H activities at Upper Lardners, the Gellibrand River at James Access and Ten Mile Creek. A similar increase in




sulphate concentrations is apparent at the Gellibrand River at James Access and at Upper

Lardners. These trends suggest increasing impacts to river water quality as a result of

anthropogenic activities within the catchments upstream of the sampling points.

## 7. Conclusions

MTTs in the six headwater catchments in the Otway Ranges vary from approximately 7 to

234 years. There are a number of uncertainties in these MTT estimates. Some, such as the

uncertainty in the rainfall $^3$H, impact all of the catchments as a whole and will thus not result

in major uncertainties in relative MTTs between catchments or within a single catchment at

different flow conditions. Likewise, uncertainty in the most suitable LPM will affect the

comparison of MTTs between catchments but not within the same catchment at different

flow conditions. Aggregation error is of a similar magnitude to many of the other

uncertainties and is more difficult to assess. Despite these uncertainties, that the MTTs are

several years to decades remains a robust conclusion. This would place them amongst the

oldest of any yet estimated globally.

The reason for the unusually long MTTs is uncertain but could be related to very low aquifer

recharge rates and/or high transpiration rates associated with eucalyptus forests (Allison et

al., 1990). The long MTTs suggest that short-term events such as drought or bushfire may

not impact the streams. However, longer-term changes within the catchments, such as land

use change, climate change or contaminant loading, may affect the streams but not for

many years. An example of this is increasing nitrate and sulphate concentrations within

several of the catchments, which implies increasing impacts to river water quality as a result

of anthropogenic activities.

There is a strong correlation between $^3$H activities and EC, major ion concentrations, and/or

TDS at Porcupine Creek and between $^3$H activities and TDS at Lardners Gauge. These

relationships allow a first order estimate of $^3$H activity and, therefore, MTTs at either of

these two locations using a single water quality measurement. More broadly, $^3$H activities

within any catchment can be estimated using a simple $^3$H-discharge relationship, which is



characterised by a discharge threshold of approximately $10^4$ $m^3$ $day^{-1}$. Despite differences in geology, catchment size, land use, drainage density, runoff, and slope, this $^3H$-discharge relationship implies that the headwater streams in the Otway Ranges behave in a relatively

uniform fashion.  This further implies that the dominating control affecting the variability in $^3H$ activities is the relative contribution of groundwater and soil water, rather than physical catchment attributes.

The $^3H$ activities of the river water samples, in combination with a correlation between MTTs and river discharge, suggest that recharge to the regional aquifer is within the range

of 0.8 to 1.5%.  These values are lower than estimates provided by previous researchers but are in line with recharge estimates made in other parts of southeast Australia.  This study demonstrates a new methodology for estimating groundwater recharge based upon $^3H$ activities in river water.

## Data Availability

All geochemistry data utilised in this study are contained in the Supplement.  River discharge data and historic EC data for Porcupine Creek are publicly available from the Victorian State Government, Department of Environment, Land, Water & Planning (DELWP), Water Measurement Information System (http://data.water.vic.gov.au/monitoring.htm)/

## Author Contributions

William Howcroft undertook the sampling program and oversaw the analysis of the geochemical parameters and the MTT calculations.  Uwe Morgenstern was responsible for the $^3H$ analysis.  The manuscript was prepared by William Howcroft, Ian Cartwright and Uwe Morgenstern.

## Acknowledgements

Field work and laboratory analyses were conducted with the help of Massimo Raveggi, Rachelle Pearson, Wang Dong, Kwadwo Osei-Bonsu and Lei Chu.  Funding for this project was provided by Monash University and the National Centre for Groundwater Research and



Training (NCGRT).  NCGRT is an Australian Government initiative supported by the

Australian Research Council and the National Water Commission via Special Research

Initiative SR0800001.



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





**Table 1**. Catchment Attributes

**Table 2**: Correlation between catchment attributes and $^3$H activities

**Table 3.** MTTs for each river water sample, as determined using the Exponential Piston Flow Model (EPM) with EPM Ratios of 0.33, 1.0 and 3.0, and the Dispersion Model (DM) with Dispersion parameters (Dp) of 0.05 and 0.5. Values in parentheses represent non-unique transit times.

**Table 4**. True and derived (sample) MTTs for Love Creek Kawarren, accounting for aggregation error.





| Catchment | Drainage Area (km²) | Drainage Density (m m⁻²) | Forest Cover (%) | Average Slope (°) | Runoff Coefficient (%) |
|---|---|---|---|---|---|
| Upper Lardners (UL) | 20.0 | $1.0 \times 10^{-3}$ | 92 | 11.0 | - |
| Lardners Gauge (LG) | 51.6 | $1.1 \times 10^{-3}$ | 91 | 11.0 | 33.0 |
| Gellibrand River (JA) | 81.0 | $9.2 \times 10^{-4}$ | 95 | 11.3 | 39.0 |
| Porcupine Creek (PC) | 33.6 | $9.5 \times 10^{-4}$ | 88 | 5.9 | 11.4 |
| Ten Mile Creek (TC) | 9.6 | $8.8 \times 10^{-4}$ | 88 | 5.7 | 12.0 |
| Yahoo Creek (YC) | 16.6 | $8.7 \times 10^{-4}$ | 95 | 8.6 | 10.5 |
| Love Creek Kawarren (LK) | 74.4 | $9.3 \times 10^{-4}$ | 82 | 6.4 | 10.6 |
| Love Creek Wonga (LW) | 91.7 | $9.2 \times 10^{-4}$ | 78 | 6.7 | 8.6 |

**Table 1**





| Catchment Attribute | Sampling Date | $R^2$ |
|---|---|---|
| Area | Jul-14 | 0.01 |
| | Sep-14 | 0.26 |
| | Mar-15 | 0.06 |
| | Sep-15 | 0.57 |
| Drainage Density | Jul-14 | 0.00 |
| | Sep-14 | 0.58 |
| | Mar-15 | 0.40 |
| | Sep-15 | 0.40 |
| Runoff Coefficient | Jul-14 | 0.10 |
| | Sep-14 | 0.66 |
| | Mar-15 | 0.94 |
| | Sep-15 | 0.19 |
| Forest Cover | Jul-14 | 0.51 |
| | Sep-14 | 0.15 |
| | Mar-15 | 0.24 |
| | Sep-15 | 0.01 |
| Slope | Jul-14 | 0.39 |
| | Sep-14 | 0.55 |
| | Mar-15 | 0.87 |
| | Sep-15 | 0.15 |

**Table 2**





| Location | Date | Q $10^3$ m$^3$ day$^{-1}$ | $^3$H (TU) | MTT (years) | | | | |
| --- | --- | --- | --- | --- | --- | --- | --- | --- |
| | | | | EPM | | | DM | |
| | | | | 0.33 | 1.0 | 3.0 | 0.05 | 0.5 |
| Upper Lardners (UL) | 10/07/2014 | - | 1.99 | 9.9 | 9.6 | 8.8 | 9.0 | 11.2 |
| | 28/09/2014 | - | 1.77 | 15.7 | 12.9 | 11.8 | 12.2 | 17.6 |
| | 20/03/2015 | - | 1.54 | 24.2 | 18.5 | (16.2, 41.4) | 16.3 | 26.2 |
| | 10/09/2015 | - | 1.99 | 8.8 | 8.2 | 8.6 | 8.3 | 9.9 |
| Lardners Gauge (LG) | 10/07/2014 | 151.3 | 1.94 | 10.8 | 10.2 | 9.3 | 9.6 | 12.3 |
| | 28/09/2014 | 32.8 | 1.94 | 10.6 | 10.1 | 9.2 | 9.5 | 12.1 |
| | 20/03/2015 | 5.0 | 1.64 | 19.8 | 15.4 | (14.1, 45.7) | 14.2 | 21.6 |
| | 10/09/2015 | 116.6 | 1.97 | 9.1 | 8.5 | 8.7 | 8.6 | 10.2 |
| | 4/11/2015 | 12.7 | 1.77 | 13.8 | 12.4 | 11.2 | 11.6 | 15.8 |
| Gellibrand River (JA) | 13/03/2012 | 18.5 | 1.90 | 15.5 | 12.3 | 11.8 | 11.7 | 17.7 |
| | 26/04/2012 | 30.4 | 1.80 | 19.2 | 14.8 | 13.1 | 13.4 | 21.4 |
| | 10/07/2014 | 255.2 | 2.04 | 8.7 | 8.7 | 8.1 | 8.2 | 9.7 |
| | 28/09/2014 | 39.1 | 1.93 | 10.8 | 10.2 | 9.4 | 9.7 | 12.4 |
| | 20/03/2015 | 8.8 | 1.73 | 16.2 | 13.5 | 12.2 | 12.6 | 18.2 |
| | 10/09/2015 | 204.4 | 2.08 | 7.3 | 6.8 | 7.7 | 7.0 | 8.1 |
| Porcupine Creek (PC) | 10/07/2014 | 50.4 | 1.97 | 10.3 | 9.8 | 9.0 | 9.2 | 11.7 |
| | 27/09/2014 | 3.3 | 1.68 | 19.3 | 14.9 | (13.9, 44.7) | 13.8 | 21.0 |
| | 20/03/2015 | 1.0 | 0.20 | 179.1 | 100.0 | 69.5 | 89.6 | 233.5 |
| | 10/09/2015 | 9.7 | 2.08 | 7.3 | 6.8 | 7.7 | 7.0 | 8.1 |
| | 4/11/2015 | 0.6 | 0.40 | 136.6 | 94.8 | 68.4 | 78.7 | 161.5 |
| Ten Mile Creek (TC) | 10/07/2014 | 8.6 | 1.74 | 17.1 | 13.6 | 12.5 | 12.7 | 18.8 |
| | 27/09/2014 | 0.6 | 1.00 | 58.3 | 68.5 | 62.5 | 60.1 | 66.3 |
| | 20/03/2015 | 0.2 | 0.44 | 128.4 | 92.5 | 67.2 | 76.4 | 149.2 |
| | 10/09/2015 | 1.7 | 1.09 | 48.3 | 55.5 | 62.0 | 57.0 | 53.5 |
| | 4/11/2015 | 0.1 | 0.53 | 109.4 | 90.3 | 67.2 | 73.3 | 130.2 |
| Yahoo Creek (YC) | 11/07/2014 | 23.0 | 2.14 | 6.9 | 6.8 | 7.2 | 7.0 | 7.6 |
| | 28/09/2014 | 1.2 | 1.19 | 44.7 | 52.0 | (60.6, 27.4) | (55.3, 24.8) | 49.2 |
| | 20/03/2015 | 0.4 | 0.43 | 132.1 | 93.1 | 67.4 | 77.2 | 153.7 |
| | 10/09/2015 | 3.9 | 1.30 | 34.8 | 31.3 | (34.3, 60.0) | (27.6, 50.7) | 37.9 |
| Love Creek Kawarren (LK) | 10/07/2014 | 102.9 | 1.85 | 13.3 | 11.5 | 10.5 | 10.9 | 15.0 |
| | 27/09/2014 | 6.7 | 1.34 | 35.3 | 33.5 | (32.3, 59.2) | (24.8, 51.2) | 38.4 |
| | 20/03/2015 | 2.0 | 0.48 | 121.4 | 91.2 | 67.0 | 75.1 | 141.1 |
| | 10/09/2015 | 18.6 | 1.91 | 10.4 | 9.8 | 9.5 | 9.5 | 11.9 |
| | 4/11/2015 | 1.2 | 0.58 | 100.3 | 88.6 | 66.8 | 71.5 | 120.4 |
| Love Creek Wonga (LW) | 10/07/2014 | 103.5 | 1.86 | 13.1 | 11.4 | 10.4 | 10.8 | 14.8 |
| | 28/09/2014 | 6.0 | 1.34 | 35.7 | 34.2 | (32.1, 59.3) | (24.8, 51.4) | 38.8 |
| | 20/03/2015 | 2.0 | 0.55 | 109.1 | 89.4 | 66.4 | 72.6 | 127.0 |
| | 10/09/2015 | 19.6 | 1.88 | 11.0 | 10.4 | 9.8 | 9.9 | 12.6 |

**Table 3**





| Sample Date | MTT, Love Creek Kawarren (years) | |
|---|---|---|
| 10/07/2014 | True MTT | 15.4 |
| | Sample MTT | 11.5 |
| | Difference (years) | 3.9 |
| | Difference (%) | 25.5 |
| 27/09/2014 | True MTT | 40.9 |
| | Sample MTT | 33.5 |
| | Difference (years) | 7.4 |
| | Difference (%) | 18.1 |
| 20/03/2015 | True MTT | 87.4 |
| | Sample MTT | 91.2 |
| | Difference (years) | 3.8 |
| | Difference (%) | 4.4 |
| 10/09/2015 | True MTT | 15.5 |
| | Sample MTT (years) | 9.8 |
| | Difference (years) | 5.7 |
| | Difference (%) | 36.7 |

**Table 4**





**Figure Captions**

**Fig. 1**. Map of study area showing catchments, sampling locations and bedrock geology. Source: DataSearch Victoria (2015). LG = Lardners Gauge, UL = Upper Lardners, JA = Gellibrand River at James Access, PC = Porcupine Creek, TC = Ten Mile Creek, YC = Yahoo Creek, LK = Love Creek Kawarren, and LW = Love Creek Wonga.

**Fig. 2.** Discharge conditions under which samples were collected relative to flow duration curves for a) Lardners Gauge, b) Gellibrand River at James Access (black circles indicate data from Atkinson (2014), c) Porcupine Creek), d) Ten Mile Creek, e) Yahoo Creek and f) Love Creek (black circle represents sample collected at Love Creek Kawarren in November 2015). Source: DELWP, 2017.

**Fig. 3.** Hydrographs showing flow conditions under which samples were collected at: a) Lardners Gauge and b) Love Creek.  Black circle represents the sample collected at Love Creek Kawarren in November 2015.  Source: DELWP, 2017.

**Fig. 4.** Tritium activity as a function of discharge for all catchments except Upper Lardners.

**Fig. 5**. $^3$H activities as a function of TDS for all catchments.

**Fig. 6**. $^3$H activity as function of a) nitrate concentrations, and b) sulphate concentrations.

**Fig. 7**. Correlation between $^3$H activities and runoff coefficients for samples collected in March 2015.

**Fig. 8**. Variation in MTTs for $^3$H activities in the river water samples ranging from 0.20 to 2.14 TU using the Exponential Piston Flow Model (EPM) with EPM ratios of 0.33, 1.0 and 3.0, and the Dispersion Model (DM) with $D_p$ values of 0.05 and 0.5.

**Fig. 9**. MTTs calculated using the EPM model with an EPM ratio of 1.0 as a function of discharge for a) Lardners Gauge (LG), b) Gellibrand River at James Access (JA) where black circles represent samples collected by Atkinson (2014), c) Porcupine Creek (PC), d) Ten Mile Creek (TC), e) Yahoo Creek (YC), and f) Love Creek, where blue circles represent Love Creek Kawarren (LK) and red circles represent Love Creek Wonga (LW).

**Fig. 10**. Variation in MTTs using the EPM model with an EPM ratio of 1.0 and variable rainfall input records.

**Fig. 11**: Variation in MTT as a function of discharge at Porcupine Creek based upon DELWP data for the period January 1990 to January 1994.





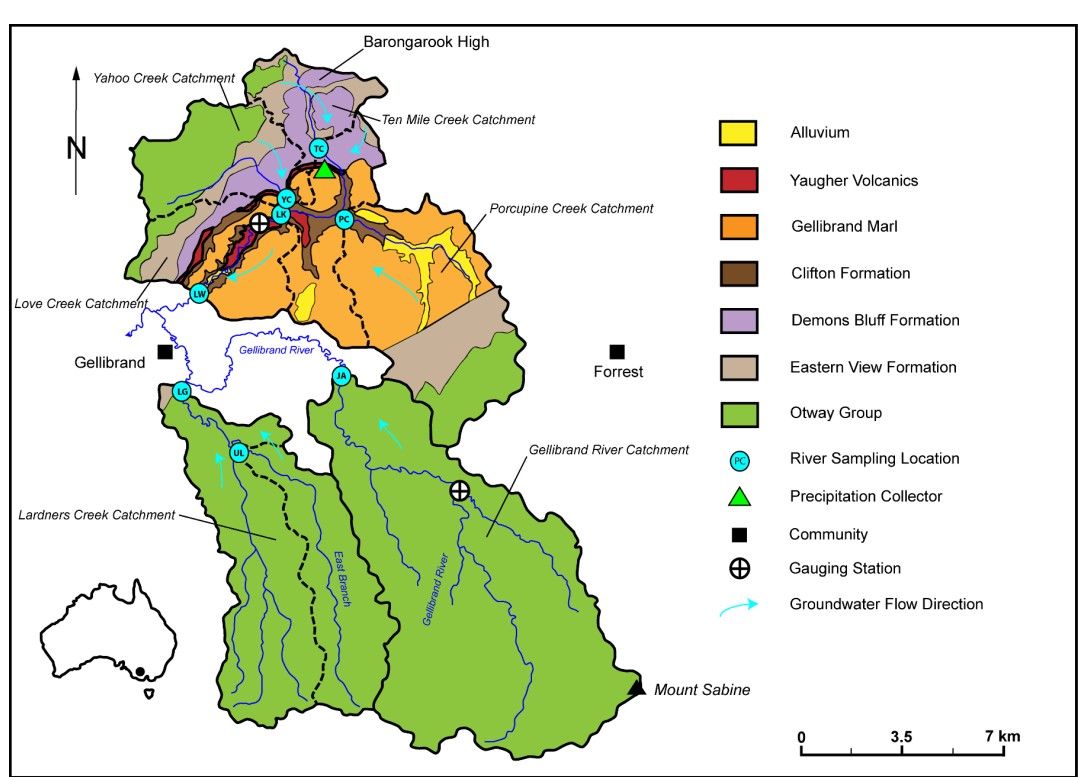

**Fig. 1**



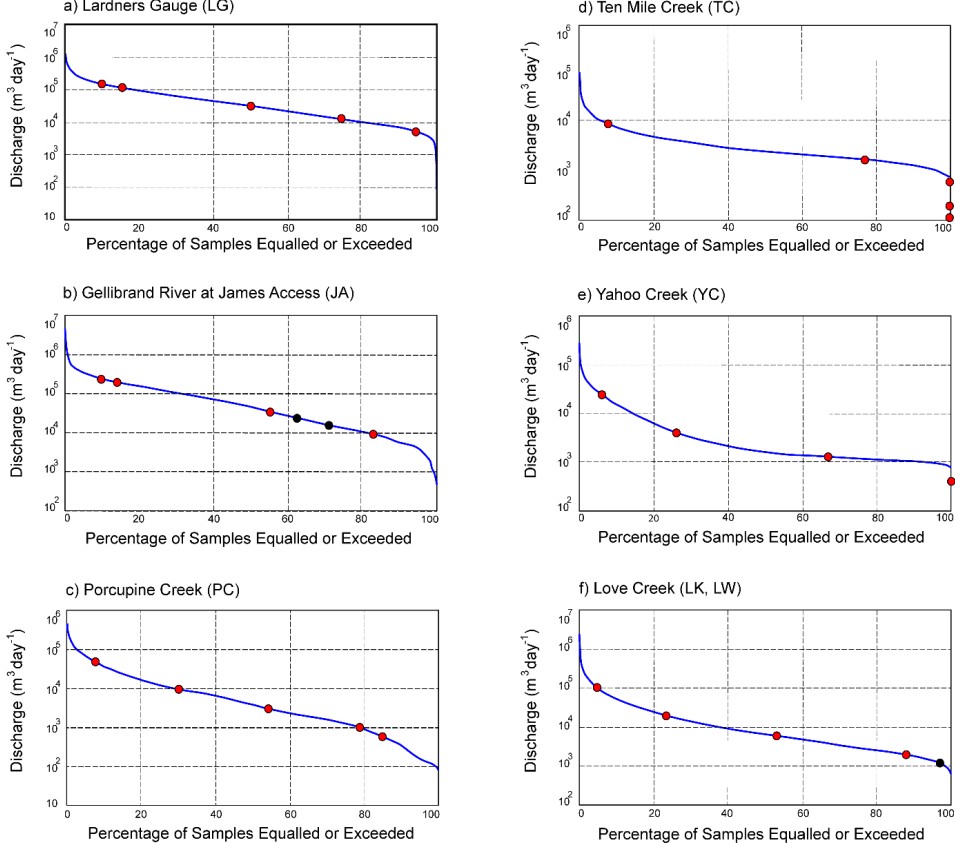

**Fig. 2**



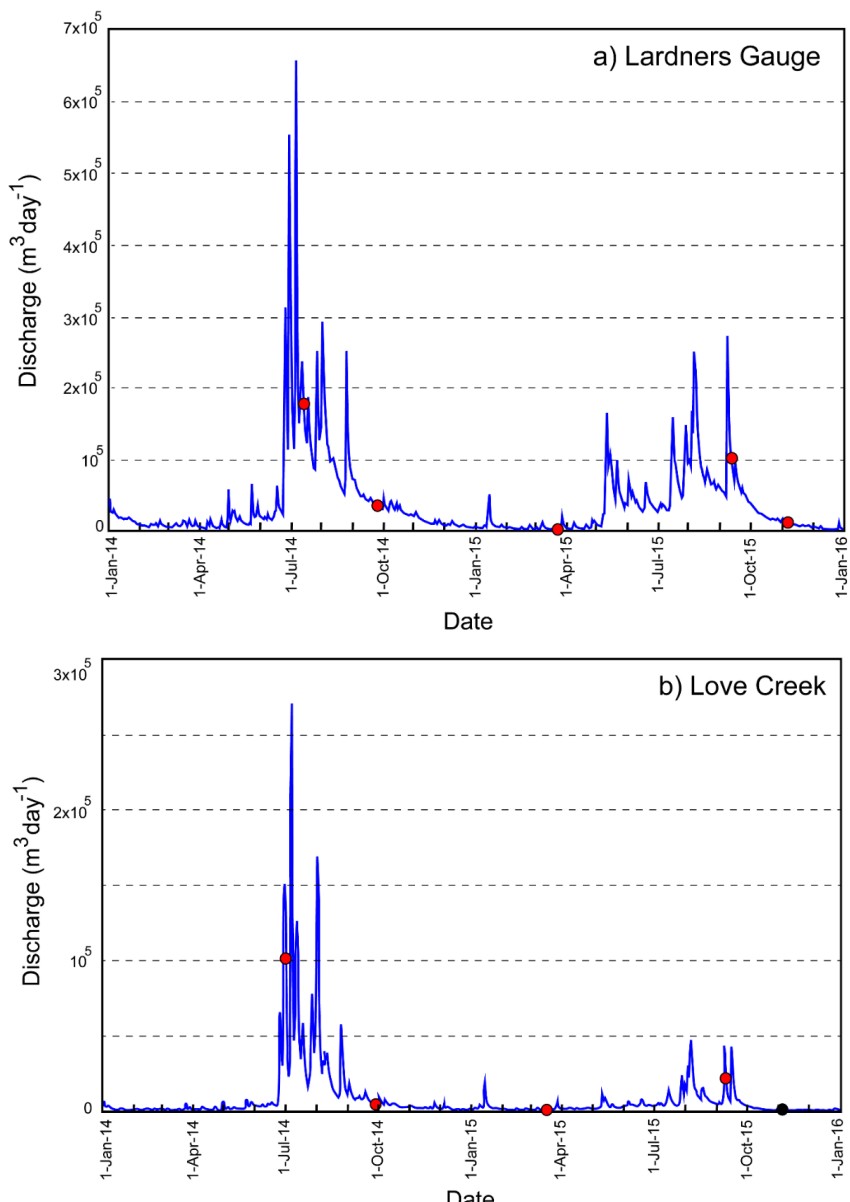

**Fig. 3**





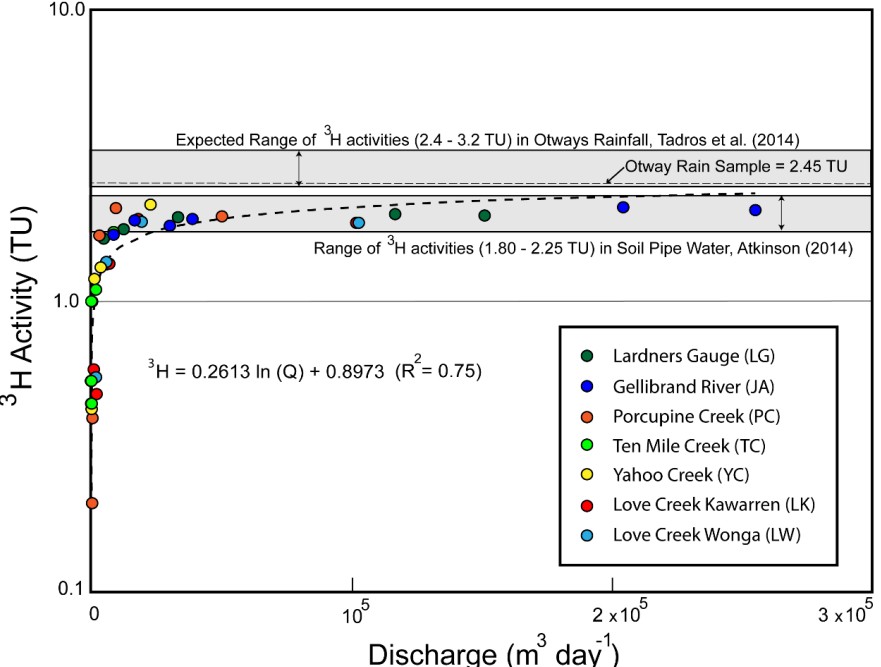

**Fig. 4**




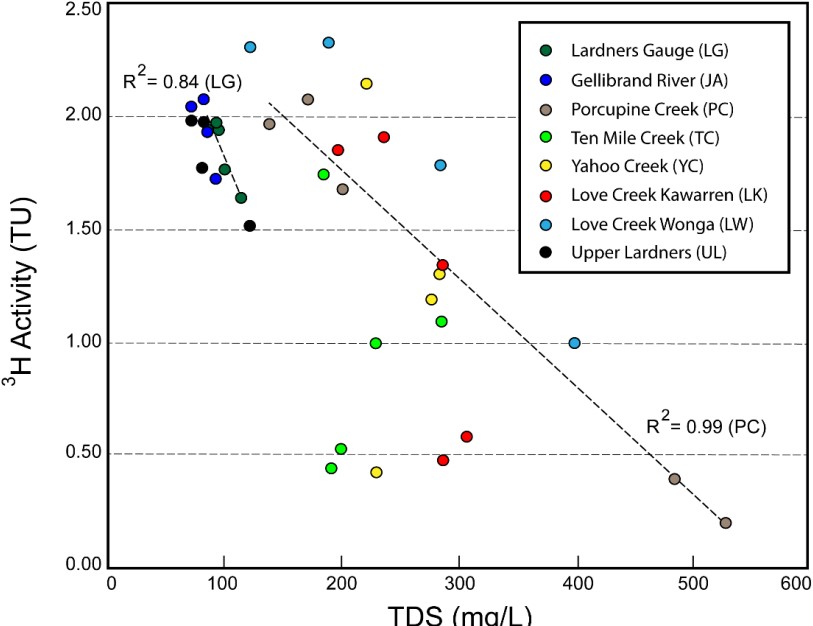

**Fig. 5**





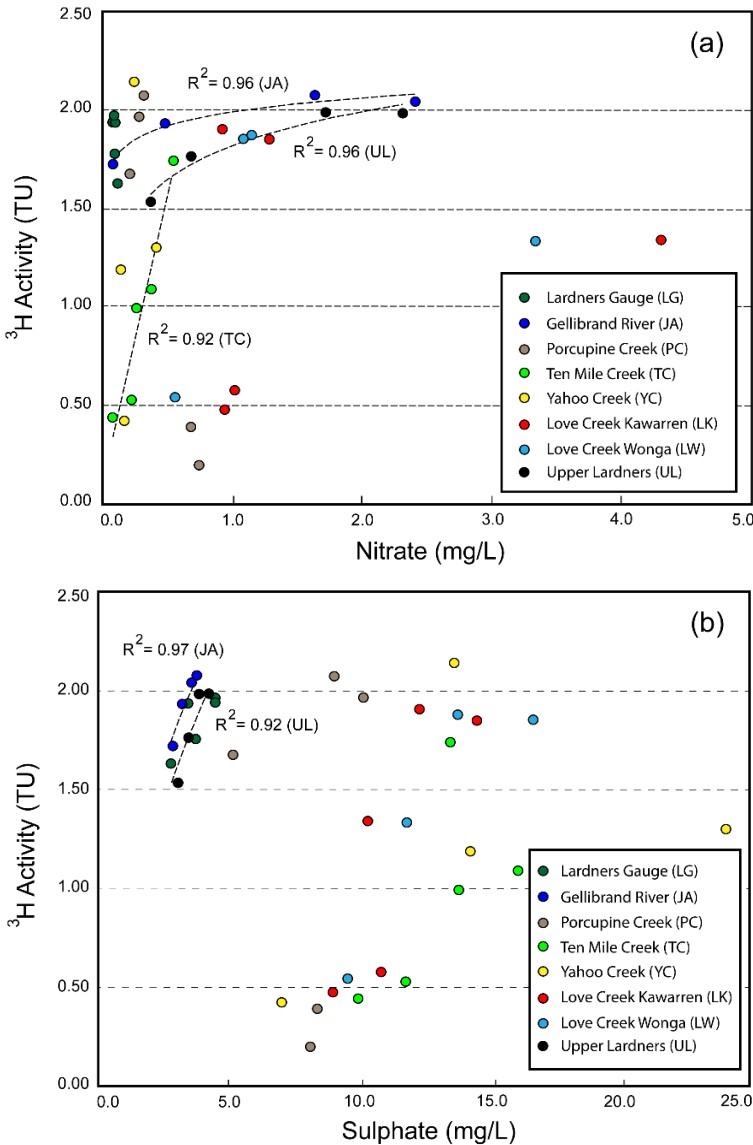

**Fig. 6**





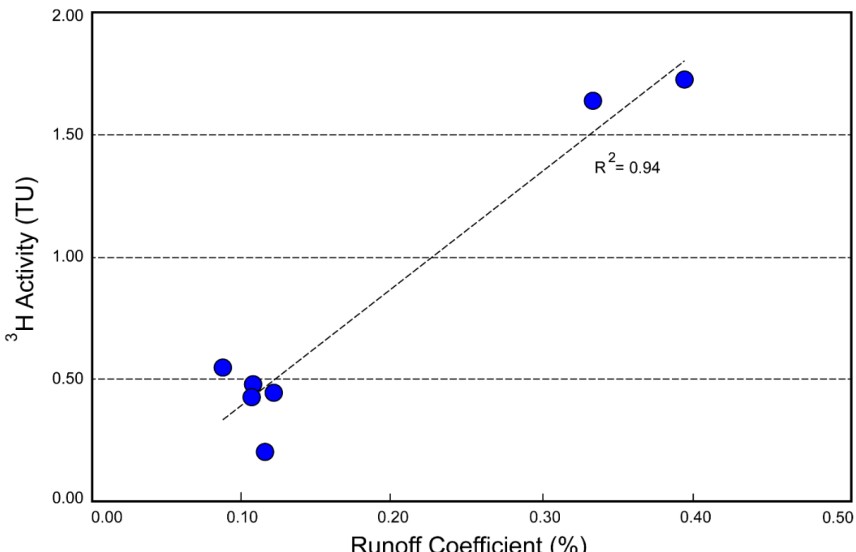

**Fig. 7**





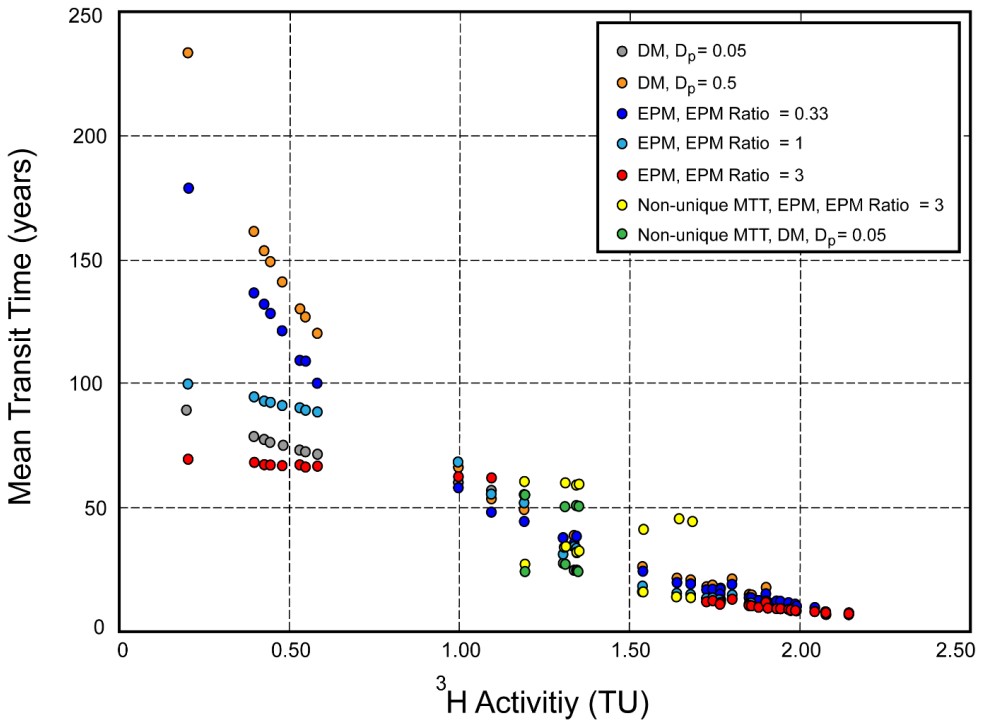

**Fig. 8**



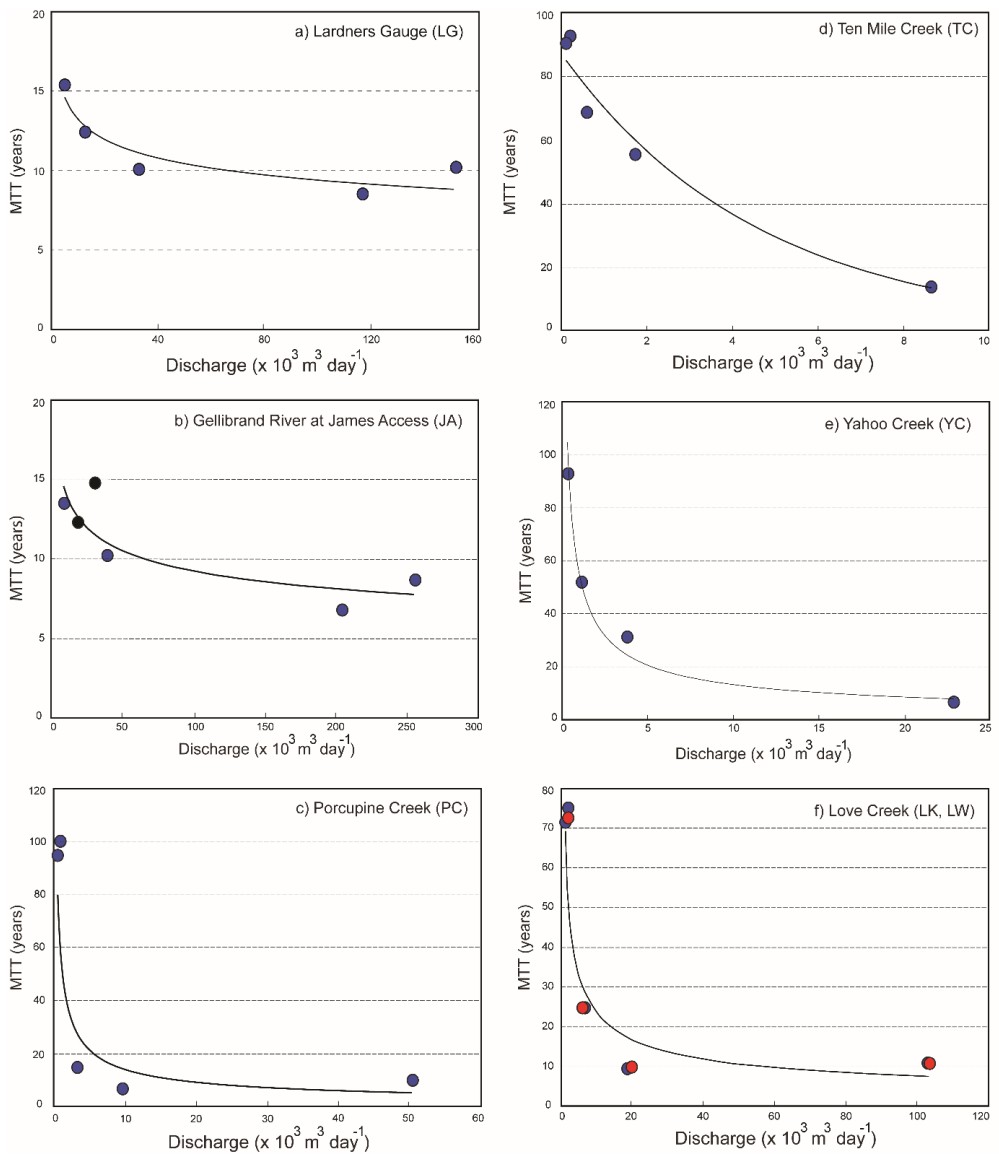

**Fig. 9**




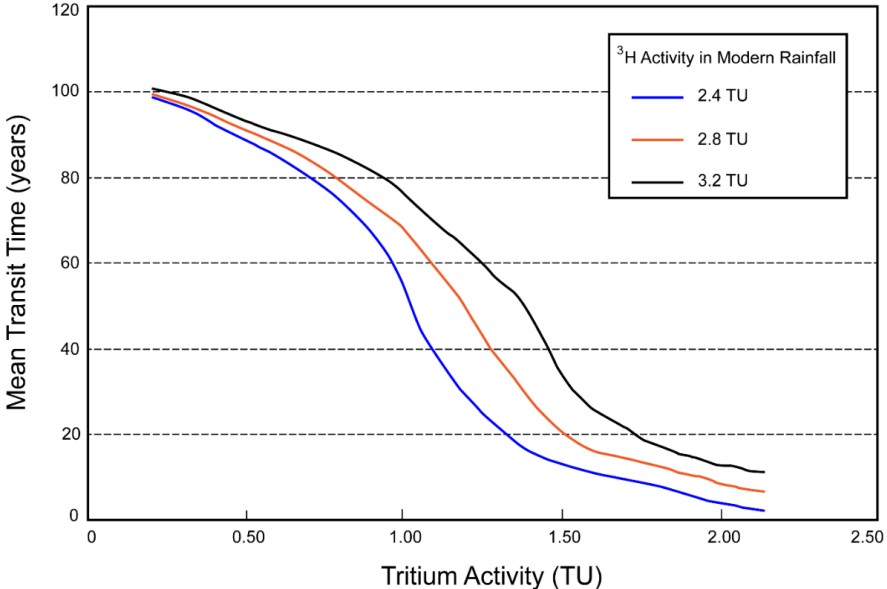

**Fig. 10**





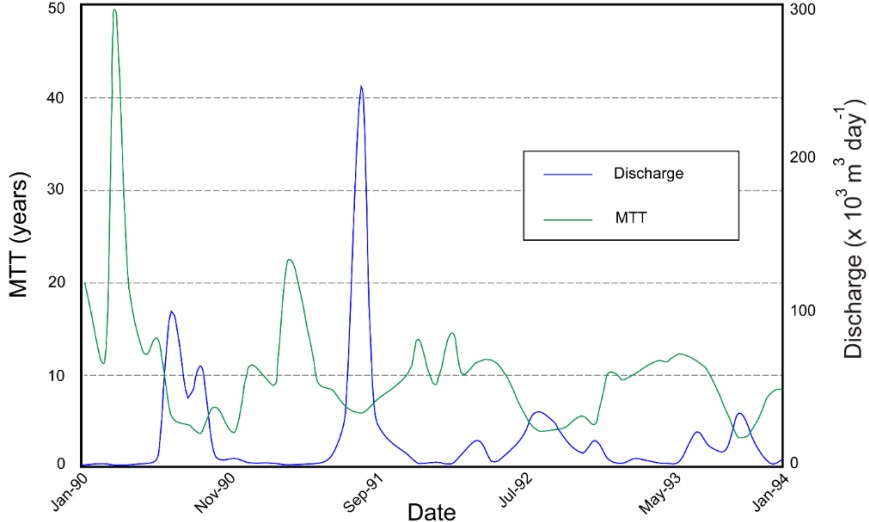

**Fig. 11**