# Peer review of "Mean Transit Times in Headwater Catchments: Insights from the Otway Ranges, Australia"

_Hydrology and Earth System Sciences, 2017_

## Referee Comment (RC1)

**Review of "Mean Transit Times in Headwater Catchments: Insights from the Otway Ranges, Australia" by Howcroft et al. (2017).**

**Summary and Recommendation**

In this paper the authors estimate the mean transit times (MTT) of six headwater catchments in south-east Australia. Their goal is to gain a better understanding of the catchment's function to enable an improved water resources management in these areas. To estimate the MTTs the authors use tritium ($^3$H) activities, major ion geochemistry and discharge data in combination with lumped parameter models (LPM). The authors discuss the uncertainties of their approach partly conclusive, try to identify possible proxies for $^3$H as well as shortly compare water quality variables with their estimated MTTs. Based on these results and the unusual long MTTs, the authors conclude that changes like droughts, deforestation and other forms of land use changes would not be realized within the streams for at least a decade.

Overall, the manuscript is well-organized. The introduction, methodology and discussion sections are well written, although sometimes I feel that restructuring some parts would improve the overall structure of the manuscript (see technical details). The results are well presented using adequate figures and tables. However, I have some concerns regarding the possible generalization and the overall connection of the results to the system functioning (see the major comments below). Nevertheless, I suggest that this paper could be accepted for publication in HESS, although I believe some major revisions are needed beforehand. I am looking forward to receiving the replies of the authors.

**Major comments**

1) *Generalizability*

Line 158: "*It is expected that the results of this investigation will facilitate greater understanding of headwater streams not only within the Otway Ranges but in similar catchments worldwide.*"

Line 626: "*This study demonstrates a new methodology for estimating groundwater recharge based upon 3H activities in river water.*"

I encourage the authors to reconsider why a potential reader from a different part of the world should read your manuscript? If as you propose in Line 158 your approach will facilitate greater understanding of headwater streams worldwide you should discuss which of your results are general and which are more specific to your landscape. Furthermore, I would reformulate your primary and secondary objective with a stronger focus on generalizability.

If your goal is to develop a novel methodology as written in Line 626 you should make that clear at the beginning of your manuscript, state a clear hypothesis and explain what is new compared to other approaches. However, if you prefer keeping your primary and secondary objectives as they are, which is perfectly valid, you should consider moving this paper to the "cutting-edge case studies", a relative new type of publication form in HESS.

2) *Model selection*

Line 117: "*As a consequence, LPMs must typically be assigned based upon knowledge of the geometry of the flow system and/or information from previous time-series studies.*"

It is interesting that you develop a perception of how you think the catchments are functioning to justify the basic assumptions of your general approach (see major comment 4) and upon which you chose your LPMs. However, in Line 464 you write that it is not possible to assess the most suitable LPM in your study which means that all chosen models are equally likely, doesn't it? Though, just in the following lines you discuss which LPMs results are more or less realistic. To avoid confusion, I think you should clarify this in your manuscript and clearly state if you can constrain your model results or not.

*3) System understanding*

Overall, I found the discussion of your MTTs results a little short with respect to your system understanding. I recommend that you discuss in more detail, if and which of your calculated MTTs are realistic in your systems. For instance, are MTTs of 200 years and an annual groundwater recharge rate of 1 % in a headwater catchment of your geology realistic, if considering the hydraulic conductivity, the mean depth and average gradient of the groundwater bodies?

Furthermore, your calculated runoff coefficients vary from 8.6 % to 39 %. This is a pretty large spectrum, especially because the catchments are within the same climate and share a similar land-use (1.3 m of mean annual rainfall / forest cover 78 -95%). Do you have an explanation for this rather strong difference in the hydrological response? Are you seeing these clear differences in the hydrological behavior also in your MTTs and what conclusions can be drawn from this? Do the basic assumptions you need to make to apply your approach (no significant dilution of groundwater inflow; see discussion point 4) also apply in the two catchments with a runoff coefficient of around 40 %?

As you treat all catchments in a similar fashion, why do you think your MTTs are so different in your catchments? Is it a result of the uncertainty in your models or are the catchments functioning differently and if so, could you identify catchment attributes which might be the reason for this

dissimilarity? For instance, the Porcupine creek and the Yahoo share a similar runoff coefficient of 11.4 and 10.5. On the 20/03/2015, you took $^3$H samples in both catchments. If you calculate the specific discharge in both catchments, it shows that they are not too dissimilar with respect to their runoff generation at that given day. However, your MTTs differ in both catchments from a maximum of 80 years (DM 0.5) to a minimum of 2 years (EPM 3.0) and this is not the only day with such high differences.

Overall, given the large differences of your results, I encourage you to connect your research results much stronger with your system architecture and check if these results fit with the knowledge you have from these landscapes. Showing that two systems act differently can be relevant, however, identifying why they act differently is much more interesting for potential readers.

4) *The basic assumption of you approach.*

Line 428: "*The flow system may therefore be viewed as a continuum that is dominated by older groundwater inflows at low flows and progressively shallower and younger stores of water (such as soil water or perched groundwater) that are mobilised during wetter periods.*"

Line 452: "Whether this reflects changes to the flow system or is due to uncertainties in the MTTs (discussed below) is not certain."

From McGuire and McDonnell (2006) which you cite in your manuscript: "*Most methods are based on early adaptations from the chemical engineering and groundwater fields (e.g., Danckwerts, 1953; Eriksson, 1958; Maloszewski and Zuber, 1982; Haas et al., 1997; Levenspiel, 1999) and may not apply in catchments where there are complex and important controlling processes like variable flow in space and time, spatially variable transmissivity, coupled vertical and lateral flow, immobile zones, and preferential flow, to name a few.*

*…*

*These simplifications include one-dimensional transport, time-invariant transit time distributions, uniform recharge, linear and steady-state input and output relations, and contribution from the entire catchment area (Turner and Barnes, 1998).*"

First of all, I would like to highlight that I am not an expert in isotope or tracer hydrology. I apologize for the following comments in advance. Nevertheless, I believe that the following questions, which came across my mind while reading your manuscript, could help readers apart from the tracer community, to better understand your approach.

Similar as it is the case in different unit hydrograph applications, your approach assumes a time invariant and linear input-output relationship of your tracers passing your catchments. However, it has been proven that catchment responses of different kinds are highly non-linear and time variant in several studies over the last 40 years. With respect to runoff predictions, it is nowadays widely accepted that concepts like the unit hydrograph will lead to unrealistic predictions on longer time scales. If we now consider your coarse sampling (3-6 observations in each catchment), the seemingly arbitrary choice of your LPMs and the corresponding parameters as well as the time frames you are working on (up to 233 year/sampling period 1.5 years), it comes to me as no surprise that your model results are so different and highlight how speculative they are.

Furthermore, in Line 428 you propose that the flow paths in your system are state dependent. You argue that you couldn't identify significant dilution of groundwater inflow by recent rainfall at the sampling time. However, you miss a detailed explanation how you came up with this fundamental conclusion. I believe, you need to have a rather good understanding of your systems to exclude that flow paths are interacting and especially when your system is switching between the two proposed states (groundwater or soil water dominated). If you have this knowledge why do you not use it to constrain your model results?

If I made some wrong conclusions here about the necessary assumptions you need to make (linearity (superposition principle) and time invariance (your filter shouldn't be time-varying on the scale your are working), I again apologize for these comments. Nevertheless, I suggest a much more comprehensive discussion of the assumptions you need to make to apply your approach in your systems and why you think they are valid on a time scale of decades.

Minor or technical comments:

Line 27 "*The MTT of this $^3$H activity is approximately ten years, which implies that changes within the catchments, including drought, deforestation, land use and/or bush fire, would not be realised within the streams for at least a decade.*"

Line 604 to 607: "*The reason for the unusually long MTTs is uncertain but could be related to very low aquifer recharge rates and/or high transpiration rates associated with eucalyptus forests (Allison et al., 1990). The long MTTs suggest that short-term events such as drought or bushfire may not impact the streams.*"

How can you exclude that the direct reaction of the stream flow to rainfall (rise of the hydrograph) is not influenced by the named land-use changes as you only analyzed your systems at times where

they produced baseflow (following your definition). I would reformulate your statement and make clear what you mean with: *"The long MTTs suggest that short-term events such as drought or bushfire may not impact the streams."*

Line 431: I do not understand this sentence. Please rephrase.

Line 436: Are the catchments in New Zealand of which you chose one of the EPM ratios of 0.33 similar to the catchments you are working in? Have you chosen the EPM ratio of 3.0 as the minimum exponential flow (25 %) on basis of a catchment property or did you just randomly pick this value?

Line 443: July 2015 instead of 2014?

Line 442 until 449: Belongs to the method section?

Line 455 until 464: Again method section?

Line 561 until 585: I recommend to rework or remove this entire section. First of all, method, result and discussion parts are entirely mixed. Furthermore, the calculations seem to be widely speculative especially because your estimated MTTs are highly uncertain (see your subsection 6.3.1). I believe a potential reader understands that properly calculated MTTs can be used to estimate the groundwater recharge.

*Section 6.6*: Either you discuss this section in more detail with references to other studies and with a relation to the processes and potential hazards or you remove this section from your manuscript.

---

## Referee Comment (RC2) · Anonymous Referee #2 · 30 Aug 2017

The paper estimates mean transit times in 6 headwater catchments in southeast Australia using two methods and radioactive 3H tracers. The study is very interesting and provides with an initial overview that stable isotope tracers cannot provide. However, I think the discussion could be more thorough and the structure of the paper be reorganized. Following I write my suggestions to improve this paper and hopefully the authors take them in the best way possible.

General comments:

1. My first comment is a general concern since it was not mentioned anywhere in the document. Are all 3H activities used on the study normalized? If it was mentioned I missed it.

[Figure]

2. I mentioned that the results from this study are a good initial overview because the authors are ignoring the seasonal variation of tritium concentration in precipitation. In Varlam et al. (2016) and Tadros et al. (2014) is shown that seasonal variation is noticeable where autumn-winter precipitation has activities half or lower than spring season precipitation. From Tadros et al. (2014): "Within the annual cycle, a clear maximum is observed in early spring between August and September and extends into summer, with the minimum concentration occurring in March/April." The values ranging between 2.4 and 3.2TU are the annual average activities, but if the actual precipitation in March/April was at least half of the measured during those 78 days in July-September, the MTT would increase so much as it did. I understand that resources are not raining and analyzing samples for $^3$H are expensive, but this should be acknowledged as a flaw of the study and probably causing overestimation of MTT in March 2015.

3. The document lacks structure. Even when there are subtitles stating "Methodology", "Study Area", "Results" and "Discussion" there are results and methods in the discussion section, as well as study area information in the results section. I will point out in more detail in the specific comments.

4. There are a lot of regressions were the only measurement for curve fit is the $r^2$, using the p-value would also add information on the data that is correlated.

Specific comments:

1. P1 L18: 2.4 to 3.2 TU is the annual average value, not the real range of activities, big difference, which might explain partially the low values obtained on the stream water.

2. Page 6 Line 148: "agricultural" I think the authors meant "agriculture" or rephrase.

3. P7 L169: Is the forest cover in Table 1 only eucalyptus? If so, reference table 1.

4. P15 L390 to P16 L410: should be in "Study Area" section.

5. P16 L411-412: What about the correlation with geology? As well as a multiple regression or a PCA?

6. P16 L413-Fig7: This is a good example where the p-value could give more information, it's easy to see there are two extreme points with higher runoff coefficient that create that "correlation", but if those two would not be there the slope of the correlation would be negative instead of positive.

7. P17 L433-P18 L475: This should be in Results, not Discussion (with few exceptions of a couple of sentences that were discussion).

8. P18 L 476: The authors could make a section called uncertainties in the "Methodology" section with a description of each of them so there is no need to explain them in the "Discussion" section.

9. P19 Eq 2: this equation goes in the "Methodology" section, not discussion. Additionally, the "d" is missing in the equation, which is correctly mentioned in the text afterwards.

10. P20 L515-520: As mentioned before, 2.45 TU is probably the high end of activity on the annual precipitation.

11. P20 L523: If 2.45 TU is on the high end, for the March calculations the precipitation should be more on the 1-1.3 TU (being conservative).

12. P20 L530-531: Yes unimportant for the surveys taken in September, partially for those in November and July, I don't think it was unimportant in March.

13. P21 L539-L544: This should be in the Results section.

14. P21 L547-549: I agree that the intermediate flow rates are important, maybe even the

15. P21 Eq 3: This equation belongs to "Methodology", not discussion.

16. P22 Eq 4 and 5: These should be in results.

17. P22 L569-582: This belongs to results.

18. P22 L588-P23 L592: Discuss why is no increase on the sulphate concentration in the Ten Mile Creek, are the anthropogenic activities different in this catchment than in the others?

Reference:

Carmen Varlam, Octavian G. Duliu, Ionut Faurescu, Irina Vagner  Denisa Faurescu (2016) Tritium time series in precipitation of Rm. Valcea, Romania, Isotopes in Environmental and Health Studies, 52:4-5, 363-369, DOI: 10.1080/10256016.2015.1114932

---

## Author Comment (AC2) · 20 Sep 2017

Dear Reviewer,

Thank you for your review comments to our article. Please find attached our response to your comments, noting that we have drafted a combined response to both reviewers.

Regards,

William Howcroft

Please also note the supplement to this comment:
https://www.hydrol-earth-syst-sci-discuss.net/hess-2017-219/hess-2017-219-AC2-supplement.pdf

---

## Author Response (AR1)

Dear Markus

Here are the combined responses and changes made following the comments of the reviewers and editor. The corrected paper with track-changes marked is also included. We have tried to incorporate all of the substantive comments.

Bill Howcroft, Ian Cartwright, Uwe Morgenstern

**Editor's comments** (response in **blue**)

From my perspective, I would be glad if you could give special attention to the following points:

(1) in the objectives section it would be good to provide an explicit science question together with a research hypothesis. As it stands now, the objectives remain somewhat vague and unspecific.

We have rewritten the Objectives section (Section 1.3, lines 135-148) so that it is clearer what are objectives are and have framed them as a hypothesis. We have also indicted how the objectives inform research elsewhere.

(2) expand the discussion of your results to tie them more into the context of existing work, to better highlight the local and general relevance and to provide a more detailed overview of the limitations of this study.

We have significantly rewritten the Conclusions (Section 6, lines 550-599) so that they concentrate on the broader implications of the study with the more area-specific findings in the Discussion section. We have also reframed the first section of the Introduction (Section 1, lines 30-55) so that it too explains more of the general points and identifies some of the important gaps in our understanding.

(3) as noted by the reviewers, several parts of the results/discussion sections should actually go either into the study site section (catchment attributes - no matter if they were only derived for this study, only the results/discussion derived from these attributes need to go into the results/discussion sections) or into to methods section (e.g. analytical uncertainty).

We did much of the reorganisation, although it does result in the catchment attributes being presented before the methods (now lines 203-209). We also put much of the detail of how the MTTs were estimated including the $^3$H input function and uncertainties in the Methods (Section 3.4). The discussion of the MTTs themselves and the uncertainties are in the Discussion (Sections 5.2 and 5.3). These are both interpretations of the data and it is also desirable that these sections follow each other and also Section 5.1.

(4) Much is, rightfully, made of aggregation errors. However, I would urgently encourage you to reflect and eventually re-think the concept of "true" MTTs. Of course, catchments do have true MTTs. However, in the presence of aggregation errors we can by no means meaningfully establish what this true MTT is. I understand your intention, but even by assessing the MTTS of smaller sub-catchments of a given catchment, these are not true MTTs, as smaller catchments are very likely also characterized by heterogeneity. What your approach does, it helps to quantify some incremental aggregation error between catchments at different scales, and which may (or may not) give some

idea into which direction the true MTT may converge. But it will never provide you a actual true MTT.

We agree and we have rephrased the sections accordingly (e.g., lines 316-325, 474-493). We have tried to be as honest as possible with our uncertainties (Sections 3.4 and 5.3). The sensitivity analysis incorporated a much greater range in $^3$H input values than is commonly accounted for. Additionally, we have used varied the parameters in the LPMs more than in many studies. Aggregation is difficult to deal with as there is no simple way to assess it. The way that we approached it may be the worst case scenario as mixing of multiple waters produces apparent MTTs that are closer to the true value than does mixing of only a few end-members. Nevertheless the original wording was probably overoptimistic.

We have tried to put some values of the impact of the uncertainties on the MTTs (lines 494-499) and have also have illustrated while all uncertainties impact the absolute estimates of MTTs, the relative differences between MTTs within catchments at different flow conditions or within a given area are less impacted (lines 500-510). Overall, especially given that aggregation only makes the waters appear younger, the waters must be several years to decades old and the correlation of MTT with streamflow is also robust conclusion. We have stressed this in the paper (lines 550-554)

**Reviewer Comments**

Our previous responses to the reviewers' comments are below in **blue**, the resultant changes are in **green**

**Response to Reviewer #1**

- Generalizability

Line 158: "*It is expected that the results of this investigation will facilitate greater understanding of headwater streams not only within the Otway Ranges but in similar catchments worldwide.*"

Line 626: "*This study demonstrates a new methodology for estimating groundwater recharge based upon 3H activities in river water.*"

I encourage the authors to reconsider why a potential reader from a different part of the world should read your manuscript? If as you propose in Line 158 your approach will facilitate greater understanding of headwater streams worldwide you should discuss which of your results are general and which are more specific to your landscape. Furthermore, I would reformulate your primary and secondary objective with a stronger focus on generalizability.

If your goal is to develop a novel methodology as written in Line 626 you should make that clear at the beginning of your manuscript, state a clear hypothesis and explain what is new compared to other approaches. However, if you prefer keeping your primary and secondary objectives as they are, which is perfectly valid, you should consider moving this paper to the "cutting-edge case studies", a relative new type of publication form in HESS.

We will revise the Objectives and Conclusions sections of the manuscript to more specifically address which of our conclusions are of relevance globally, and which apply specifically to the Otways region of Australia. Notwithstanding numerous studies over recent years, there is still not a complete understanding of the range of mean transit times (MTT) in headwater catchments nor what controls these. The realisation that MTTs in some Australian catchments are long (years to

decades) is significant in their management, which is of local importance. That the MTTs are longer in these catchments than is perhaps commonly recorded elsewhere is important for understanding catchment behaviour more generally and we can emphasise this.

The paper has been restructured to emphasise the more general aspects and separate these from the area-specific conclusions. In particular, we have:

- Emphasised some of the current gaps in knowledge (such as the range of MTTs in headwater catchments globally, and the controls on MTTs) in the first paragraphs of the introduction (lines 30-55)
- Additionally, we have clarified the reasons that understanding MTTs is important (lines 30-48).
- Rewritten the conclusions (Section 6) so that it focusses on the broader outcomes of the study and better articulates how this study relates to our overall understanding of how headwater catchments behave.
- Rephrased the objectives (Section 1.3) so they address specific hypotheses and show how these relate to the broader understanding of the behaviour of headwater catchments.

Demonstrating a new method to estimate groundwater recharge was not a specific goal of this study. Nonetheless, through examination of groundwater volumes across different times of the year, we realised that our data could be used to estimate groundwater recharge to the regional aquifer. We are keen to retain this section as it is novel and potentially of interest to researchers elsewhere. However, it is a relatively minor part of the paper and requires testing in other regions. We chose to include this topic in our manuscript to demonstrate a broader use of MTT estimates. Given that, we consider that this paper constitutes a regular research paper. Further work in this field may be suitable for a "cutting-edge case study" type of publication.

In the end we took that recharge estimate out as it was overly speculative. For some of the catchments it was possible to estimate the turnover volumes of groundwater (which is more conventionally done) (Sections 3.5, 5.5)

- Model Selection

Line 117: "*As a consequence, LPMs must typically be assigned based upon knowledge of the geometry of the flow system and/or information from previous time-series studies.*"

It is interesting that you develop a perception of how you think the catchments are functioning to justify the basic assumptions of your general approach (see major comment 4) and upon which you chose your LPMs. However, in Line 464 you write that it is not possible to assess the most suitable LPM in your study which means that all chosen models are equally likely, doesn't it? Though, just in the following lines you discuss which LPMs results are more or less realistic. To avoid confusion, I think you should clarify this in your manuscript and clearly state if you can constrain your model results or not.

We have explained this using lines 83-93

It is true that using the approach where MTTs are estimated from individual $^3$H activities that one has to assume an appropriate LPM. The potential advantage of using $^3$H as a tracer in the southern hemisphere is that it may be used in a similar way to other radioisotope tracers (e.g. $^{14}$C or $^{36}$Cl), whereby an age or mean transit time estimate can be derived from individual measurements. In turn, this permits estimation of MTTs at a range of flow conditions.

In the northern hemisphere, the use of $^3$H as a tracer requires time series data collected over several years to estimate MTTs (due to the much larger bomb-pulse $^3$H signal). Where samples are collected at similar flow conditions (e.g. summer low flows), this permits an independent assessment of the LPM via comparison of the measured and predicted $^3$H activities. It is questionable, however, whether one could still apply the same time series approach in the southern hemisphere due to the diminution of the relic bomb-pulse $^3$H activities. For example, the calculated decrease of $^3$H activities for a water with a mean transit time of 10 years between 2016 and 2026, as predicted by the EPM and DM models used in the paper with the Melbourne $^3$H record, is only 0.2 TU. Additionally, the time vs. $^3$H trends produced by the LPMs in the southern hemisphere are similar within analytical uncertainty. Further, given the long MTTs in many southeast Australian catchments, it is not feasible to use other tracers (such as the stable isotopes) to better constrain the LPMs due to initial geochemical variations being attenuated when MTTs are more than a few years (e.g., Stewart et al., 2010, Hydrol. Process., 24, 1646-1659). We addressed these points on lines 96 to 118 of the paper and we will provide a few more details based on the above discussion to clarify our approach.

Our approach was to utilise different LPMs to bracket the estimates of MTTs. These LPMs have been
Lines 93-100 & 110-114 explain this
used in many other studies, and where time-series $^3$H data are available, they do reproduce the observed variation in $^3$H activities. The LPMs are always a simplification; however, their geometries do agree with the likely form of the flow system and thus the approach is defensible. Estimating precise MTTs is difficult and the not knowing the best LPM to apply represents an uncertainty in these calculations (as we discuss in Section 6.3). However, the conclusions that the water in these streams has MTTs of several years to decades is independent of the LPM that is employed (as was emphasised throughout the paper).

Sections 5.2 and 5.3 now discuss this more explicitly.

As a general point, with the diminishing of the bomb pulse tritium signal, MTTs in the southern hemisphere are not overly sensitive to the models. Because of this, a change in the age distribution that occurs when different LPMs are used do not change the MTT dramatically.  In the northern hemisphere, with significant bomb tritium still present, a change in the age distribution significantly changes the fraction of bomb-pulse tritium in the sample and therefore result in a different MTT.
This is explained in Section 5.2 (lines 429-497) and Section 5.3 attempts to quantify the resultant
uncertainties associated with the comparison of the MMTs from the different LPMs in section 6.2.
This is noted on line 99.

- System Understanding

Overall, I found the discussion of your MTTs results a little short with respect to your system understanding. I recommend that you discuss in more detail, if and which of your calculated MTTs are realistic in your systems. For instance, are MTTs of 200 years and an annual groundwater recharge rate of 1 % in a headwater catchment of your geology realistic, if considering the hydraulic conductivity, the mean depth and average gradient of the groundwater bodies?

The 200 year value is an absolute maximum and is subject to considerable uncertainty (as we discuss in section 6.3). However, as outlined in the response to other comments below, the conclusion that mean transit times are years to decades is robust. The long mean transit times do imply slow recharge rates. There are only sparse measurements of hydraulic conductivities in these aquifers and, consequently, it is difficult to corroborate the recharge rates using the aquifer properties. The recharge rates are consistent with those generally proposed for eucalyptus forest areas in SE

Australia. For example, Allison & Hughes (1983. Journal of Hydrology 60, 157–173), Allison et al. (1990. Journal of Hydrology 119, 1-20), Herczeg et al. (2000. Marine and Freshwater Research 52, 41-52), and Cartwright et al. (2006, Journal of Hydrology 332, 69-92) estimate that recharge rates in areas dominated by native forest are at most a few mm per year and often less. These low recharge rates are due to the high transpiration rates in eucalypt dominated catchments. Further, because hydraulic conductivities are poorly known in most areas, it is important to find other means to estimate groundwater recharge. However, we recognise that there is potential for groundwater discharge from the catchment via deeper groundwater flow pathways. If this is the case, our estimates would underestimate the true recharge rate because the proposed method accounts only for the discharge at the stream gauge, not total discharge. We will discuss these uncertainties in more detail within the revised paper.

Additionally, the lack of significant near-river alluvial sediments may be a reason why the estimated MTTs are so long. The lack of near-river alluvial sediments precludes the possibility of significant bank storage and return flow contributing to total river discharge and, thus, probably influence the MTTs. We will also discuss this in the revised paper.

As discussed earlier, we removed the original recharge rate calculations from the paper. However, low recharge rates are consistent with previous studies (e.g. Allison et al., 1990) and are common in southeast Australia (lines 553-558).

We have also improved our discussion of the uncertainties in the revised paper (Section 5.3 and Section 6, lines 550-553) and have explained more clearly that despite the uncertainties in the MTT calculations, the observation that the $^3$H activities are locally 10% of modern rainfall (and much less relative to the bomb-pulse rainfall) necessitates MTTs that are several decades (lines 500-503).

Furthermore, your calculated runoff coefficients vary from 8.6 % to 39 %. This is a pretty large spectrum, especially because the catchments are within the same climate and share a similar land-use (1.3 m of mean annual rainfall / forest cover 78 -95%). Do you have an explanation for this rather strong difference in the hydrological response? Are you seeing these clear differences in the hydrological behaviour also in your MTTs and what conclusions can be drawn from this? Do the basic assumptions you need to make to apply your approach (no significant dilution of groundwater inflow; see discussion point 4) also apply in the two catchments with a runoff coefficient of around 40 %?

The calculation of the runoff coefficient in a region with well measured rainfall and long streamflow records is relatively straightforward. What is less clear are the reasons for the variation. The high runoff coefficients for Upper Lardners, Lardners Gauge and James Access may be because these three rivers drain steeper catchments and are underlain almost entirely by low hydraulic conductivity Otway Group basement rocks.

This is now discussed on lines 342-347.

The runoff coefficients do correlate well with $^3$H activities and the reason that we included them in this study is that they are useful in providing a first-order estimate of MTTs (in as much as they indicate whether the water is likely to be relatively young or old). The variation in the runoff coefficients is probably controlled by similar factors that control the variation in the MTTs. Catchments with low recharge rates may lose water to the atmosphere by evapotranspiration and consequently have both long MTTs and low runoff coefficients. However, as noted in Section 5.4, it is unclear whether and how catchment attributes such as slope, drainage density control the MTTs (and so by extension the runoff coefficients). A lack of a single catchment attribute controlling MTTs

was also noted by Cartwright & Morgenstern (2015. Hydrol. Earth Syst. Sci., 20, 4757-4773) in the Ovens catchment of NE Victoria. In that catchment there was also a good correlation between the [3]H activities and the runoff coefficient. We will explain the importance of the correlation in terms of providing first order estimates of MTTs in the revised manuscript.

We have clarified the correlation of [3]H (and by extension MTTs) with the runoff coefficient (lines 524-529). We have explained that both are probably controlled by recharge and groundwater flow rates and that the correlation is probably useful in estimating broad differences in MTTs rather than more subtle differences. The general importance of using the runoff coefficient to estimate MTTs is also highlighted (lines 592-595)

As you treat all catchments in a similar fashion, why do you think your MTTs are so different in your catchments? Is it a result of the uncertainty in your models or are the catchments functioning differently and if so, could you identify catchment attributes which might be the reason for this dissimilarity? For instance, the Porcupine creek and the Yahoo share a similar runoff coefficient of 11.4 and 10.5. On the 20/03/2015, you took 3H samples in both catchments. If you calculate the specific discharge in both catchments, it shows that they are not too dissimilar with respect to their runoff generation at that given day. However, your MTTs differ in both catchments from a maximum of 80 years (DM 0.5) to a minimum of 2 years (EPM 3.0) and this is not the only day with such high differences.

The causes of the variations of the MTTs between the catchments remains an open question. In studies elsewhere, catchment attributes such as slope and drainage density were shown to correlate with MTTs. Given the multiple interacting processes that control the transmission of water through catchments (e.g., as discussed by McGuire and McDonnell, 2006; Hrachowitz et al., 2009; Stewart & Fahey, 2010), it is probably not surprising that no single catchment attribute controls mean transit times. Moreover, the lack of correlation confirms that multiple processes control water flux, and that these processes and their interaction are still poorly understood. Similar variations in MTTs between streams are also apparent in the Ovens Catchment (Cartwright & Morgenstern, 2015) and, in that case, the reasons are also not clear. While it is a negative outcome, it is worth us emphasising the lack of correlation between [3]H and the catchment attributes in the Discussion section, as it is important.

We have clarified the discussion around not being readily able to determine the controls on the MTTs both in this region and in general (Section 5.4, Section 6 lines 559-576). We have also highlighted this as one of the key gaps in regionalising MTTs in upper catchments (lines 145-147, 596-599).

Overall, given the large differences of your results, I encourage you to connect your research results much stronger with your system architecture and check if these results fit with the knowledge you have from these landscapes. Showing that two systems act differently can be relevant, however, identifying why they act differently is much more interesting for potential readers.

We agree that we can better integrate the results. As we discuss in response to later comments, while the calculation of MTTs has uncertainties (many of which we discuss in Section 6.3), the observation that the [3]H activities are far below those of modern rainfall means that the MTTs must be several years to decades. The reasons for the variations in MTTs are not clear, but making that observation is also of general importance.

As above, we have clarified and expanded the discussion the reasons that MTTs differ within these catchments and more generally (Section 5.4, Section 6 lines 559-576).

- The basic assumptions of your approach

Line 428: "*The flow system may therefore be viewed as a continuum that is dominated by older groundwater inflows at low flows and progressively shallower and younger stores of water (such as soil water or perched groundwater) that are mobilised during wetter periods.*"

Line 452: "Whether this reflects changes to the flow system or is due to uncertainties in the MTTs (discussed below) is not certain."

From McGuire and McDonnell (2006) which you cite in your manuscript: "*Most methods are based on early adaptations from the chemical engineering and groundwater fields (e.g., Danckwerts, 1953; Eriksson, 1958; Maloszewski and Zuber, 1982; Haas et al., 1997; Levenspiel, 1999) and may not apply in catchments where there are complex and important controlling processes like variable flow in space and time, spatially variable transmissivity, coupled vertical and lateral flow, immobile zones, and preferential flow, to name a few. These simplifications include one-dimensional transport, time-invariant transit time distributions, uniform recharge, linear and steady-state input and output relations, and contribution from the entire catchment area (Turner and Barnes, 1998).*"

We agree the LPM models are an approximation of real-world situations. Nevertheless, they are commonly used and have successfully predicted variation in tracer concentrations / activities in many catchments. It is generally not possible to constrain all the variations in hydraulic properties in a catchment and all modelling approaches contain some elements of generalisation.

The assumption regarding time-invariance is only correct where mean transit times are calculated from time series measurements (of $^3$H or other tracers). Because $^3$H is radioactive, it will yield a mean transit time regardless of whether the catchment is time invariant as long as the flow path geometry remains relatively constant. Further, there is no requirement that water from the entire catchment reaches the stream. The much-used exponential-piston flow model, for example, is applicable to catchments that have both confined and unconfined portions.

Regardless of the uncertainties in the LPM calculations, one can get a general idea of timescales from the $^3$H activities. If a water with a $^3$H activity of modern rainfall (~2.7 TU) were collected and isolated, it would take 30 to 40 years for the $^3$H to decay to the lowest $^3$H activities recorded in the streams (0.2 to 0.5 TU). Given that the $^3$H activity of rainfall in the past 50 years was considerably higher, the timescales would be even longer. This is not a real calculation of water age or MTT; however it highlights that $^3$H is an important qualitative or semi-quantitative tracer over and above its use in the calculations (i.e. waters with low $^3$H activities are relatively old).

Some of this discussion is in the current version of the paper and we can expand on these points as they are important and perhaps not clear to a broad readership.

We have clarified these points in the revised paper.
- The discussion of uncertainties in MTTs is now more explicit (Sections 3.3 and 5.3)
- The section on LPMs and MTTs in the introduction has been revised to recognise the simplification of LPMs (however, it is still the case that they are probably a more viable alternative to estimating MTTs than alternative methods). Lines 57-65.
- The point that the low $^3$H activities necessitates long MTTs despite any uncertainties in the calculations is also made explicitly (lines 500-502).

First of all, I would like to highlight that I am not an expert in isotope or tracer hydrology. I apologize for the following comments in advance. Nevertheless, I believe that the following questions, which

came across my mind while reading your manuscript, could help readers apart from the tracer community, to better understand your approach.

Similar as it is the case in different unit hydrograph applications, your approach assumes a time invariant and linear input-output relationship of your tracers passing your catchments. However, it has been proven that catchment responses of different kinds are highly non-linear and time variant in several studies over the last 40 years. With respect to runoff predictions, it is nowadays widely accepted that concepts like the unit hydrograph will lead to unrealistic predictions on longer time scales. If we now consider your coarse sampling (3-6 observations in each catchment), the seemingly arbitrary choice of your LPMs and the corresponding parameters as well as the time frames you are working on (up to 233 year/sampling period 1.5 years), it comes to me as no surprise that your model results are so different and highlight how speculative they are.

As noted above, the use of $^3$H does not assume time-invariance. Also, because the $^3$H activities in the streams are much lower than those of rainfall, the conclusions that the MTTs must be years to decades are robust. We have been clear throughout the paper that there are considerable uncertainties in the calculated MTTs and have sought to address these where possible. However, the data allows an understanding of the broad mean transit times between and within the catchments, which was the main objective.

We have made our discussion on uncertainties clearer and emphasised what may be concluded with more certainty (the fact that MTTs must be years to decades and the relative differences between different flow conditions in the same catchment). We have also noted the point regarding time-invariance (lines 113-114).

Furthermore, in Line 428 you propose that the flow paths in your system are state dependent. You argue that you couldn't identify significant dilution of groundwater inflow by recent rainfall at the sampling time. However, you miss a detailed explanation how you came up with this fundamental conclusion. I believe, you need to have a rather good understanding of your systems to exclude that flow paths are interacting and especially when your system is switching between the two proposed states (groundwater or soil water dominated). If you have this knowledge why do you not use it to constrain your model results?

The conclusion comes from a variety of observations. Firstly, although we sampled throughout the year and at different flows, we avoided sampling immediately after heavy rainfall when new or event water may be important. Secondly, at the time of sampling, the major ion concentrations in the river do not suggest that there has been dilution between low salinity recent rainfall and older water from within the catchment. The observation that the $^3$H activities appear to plateau at values that are less than those of rainfall also implies that, during our sampling, the rivers were not dominated by recent rainfall. Finally, during the sampling rounds, there was no overland flow observed in the catchments. Our conceptualisation is that the catchment contains several stores of water ranging from deeper groundwater to shallower soil water that progressively become more important as the catchment "wets up" during the winter months. The observation that the highest$^3$H activities are similar to those recorded in soil / regolith water in this catchment is also consistent with that idea. Section 6.1 discusses this and we can add some of the above details to explain our reasons more fully.

We have revised Section 5.1 to clarify these points.

If I made some wrong conclusions here about the necessary assumptions you need to make (linearity (superposition principle) and time invariance (your filter shouldn't be time-varying on the

scale you are working), I again apologize for these comments. Nevertheless, I suggest a much more comprehensive discussion of the assumptions you need to make to apply your approach in your systems and why you think they are valid on a time scale of decades.

The comments were valuable as many papers are written from a point of assuming a high level of background knowledge. Without turning the paper into a review article, we will broaden the explanation of these issues.

- Minor or technical comments

Line 27 "*The MTT of this $^3$H activity is approximately ten years, which implies that changes within the catchments, including drought, deforestation, land use and/or bush fire, would not be realised within the streams for at least a decade.*"

Line 604 to 607: "*The reason for the unusually long MTTs is uncertain but could be related to very low aquifer recharge rates and/or high transpiration rates associated with eucalyptus forests (Allison et al., 1990). The long MTTs suggest that short-term events such as drought or bushfire may not impact the streams.*"

How can you exclude that the direct reaction of the stream flow to rainfall (rise of the hydrograph) is not influenced by the named land-use changes as you only analyzed your systems at times where they produced baseflow (following your definition). I would reformulate your statement and make clear what you mean with: "*The long MTTs suggest that short-term events such as drought or bushfire may not impact the streams.*"

That is what we meant to say and we will clarify this in the revised paper. The long MTTs mean that the base flows in the streams are buffered against short-term variations in rainfall (and indeed many of these streams continued to flow through the Millennium drought between 1996 and 2009) but that longer-term climate change will probably impact the catchments.

This is reworded in the Introduction (lines 32-38) and Conclusions (lines 577-584).

Line 431: I do not understand this sentence. Please rephrase.

This sentence notes that if the system does contain more than a single store of water (e.g. old baseflow and young event water), then the calculated MTT gives the minimum age of the baseflow component. We will rephrase it.

We have clarified this statement (lines 417-419).

Line 436: Are the catchments in New Zealand of which you chose one of the EPM ratios of 0.33 similar to the catchments you are working in? Have you chosen the EPM ratio of 3.0 as the minimum exponential flow (25 %) on basis of a catchment property or did you just randomly pick this value?

The catchments have a broadly similar geometry to those in New Zealand. The flow system here and in those examples comprises an unsaturated zone overlying the aquifers which is the basis for the choice of the EPM model (piston flow through the unsaturated zone followed by exponential flow through the aquifer). In recognition that we cannot constrain the suitable LMP, we utilised a range of values for the EPM ratio. An EPM ratio of 3 is a system with 75% piston flow. This may be too high in reality, but it does help limit the calculated range of MTTs.

We have reworded this section to indicate that the range of parameters that we have used is based on catchments elsewhere with similar geometries where time-series data are available (lines 421-425).

Line 443: July 2015 instead of 2014?

The date should be July 2014; we will correct this.

Corrected

Line 442 until 449: Belongs to the method section?

Respectfully, we disagree. Lines 442-449 discuss the MTT results, not how the MTTs were derived. Consequently, we will keep this in the discussion.

Line 455 until 464: Again method section?

As above, this paragraph does not discuss the methodology, but is a discussion of the MTT results. Again, we will keep in this section.

We have reorganised the paper to take into account the comments of both reviewers and editor. Specifically:
- All details of how the MTT calculations were made are now in the Methods as is the explanation of the uncertainties (Section 3.4)
- The actual MTT calculations are in the Discussion (Section 5.2).
- The details of the uncertainties are in the Discussion (Section 5.3).
This makes it clear what is background information (Methods) and what is interpretation (Discussion).

Line 561 until 585: I recommend to rework or remove this entire section. First of all, method, result and discussion parts are entirely mixed. Furthermore, the calculations seem to be widely speculative especially because your estimated MTTs are highly uncertain (see your subsection 6.3.1). I believe a potential reader understands that properly calculated MTTs can be used to estimate the groundwater recharge.

As discussed earlier, this was an unintended but nevertheless important finding of this study. This was removed Estimating groundwater recharge is difficult and we have proposed a way to estimate it from the MTTs. Estimating groundwater recharge from groundwater MTTs is common; however, we are unaware of anyone attempting it from the MTTs of river water.

Section 6.6: Either you discuss this section in more detail with references to other studies and with a relation to the processes and potential hazards or you remove this section from your manuscript.

It is acknowledged that this section is a minor component of the study. Nonetheless, the data suggest that anthropogenic impacts to several of the streams have occurred and, for this reason alone, is worth mentioning. On a more global scale, these data demonstrate the usefulness of using $^3$H in water quality studies, much in the way that Morgenstern and Daughney (2012) used $^3$H activities to assess baseline groundwater quality in New Zealand. We will add more detail in the revised paper.

We integrated this material into the other sections. Specifically, the observations that nitrate correlates with $^3$H and streamflow are discussed on lines 410-415 and the possible implications of this are on lines 580-585.

**Response to Reviewer #2**

The paper estimates mean transit times in 6 headwater catchments in southeast Australia using two methods and radioactive $^3$H tracers. The study is very interesting and provides with an initial overview that stable isotope tracers cannot provide. However, I think the discussion could be more thorough and the structure of the paper be reorganized. Following I write my suggestions to improve this paper and hopefully the authors take them in the best way possible.

We are grateful for the suggestions.

General comments:

1. My first comment is a general concern since it was not mentioned anywhere in the document. Are all 3H activities used on the study normalized? If it was mentioned I missed it.

The $^3$H activities are absolute values measured against the NIST standard. This is described by Morgenstern and Taylor (2009), which we cite in section 4.2, but we can add this detail to the paper.

This is noted on line 259 as is the definition of a TU

2. I mentioned that the results from this study are a good initial overview because the authors are ignoring the seasonal variation of tritium concentration in precipitation. In Varlam et al. (2016) and Tadros et al. (2014) is shown that seasonal variation is noticeable where autumn-winter precipitation has activities half or lower than spring season precipitation.  From Tadros et al. (2014): "Within the annual cycle, a clear maximum is observed in early spring between August and September and extends into summer, with the minimum concentration occurring in March/April." The values ranging between 2.4 and 3.2TU are the annual average activities, but if the actual precipitation in March/April was at least half of the measured during those 78 days in July-September, the MTT would increase so much as it did. I understand that resources are not raining and analyzing samples for 3H are expensive, but this should be acknowledged as a flaw of the study and probably causing overestimation of MTT in March 2015.

It is true that $^3$H activities in rainfall have seasonal variation.  However, for waters with long mean transit times, this has little impact on the calculated MTTs unless recharge occurs dominantly during periods when rainfall either has high or low $^3$H (see discussion in Morgenstern et al., 2010 doi: 10.5194/hess-14-2289-2010). The Otways have high rainfall distributed through the year and consequently there is not a distinct recharge season. The seasonal variation of $^3$H activities in SE Australia is ~1 TU which is similar to the range of $^3$H activities that we calculated MTTs for (2.4 to 3.2 TU) and so any potential impact of seasonal recharge is likely to be a similar order of magnitude or less. It does not alter the overall conclusions that MTTs are years to decades. We will include some discussion pertaining to this issue in our revised manuscript.

This has been explained in detail (lines 463-474).

3. The document lacks structure. Even when there are subtitles stating "Methodology", "Study Area", "Results" and "Discussion" there are results and methods in the discussion section, as well as study area information in the results section. I will point out in more detail in the specific comments.

We will ensure that the material in in the correct section.  We address the reviewer's specific comments that relate to the structure of the paper below. However, in many cases, we consider that we have the material in the correct sections. Specifically, our results section presented the data and the discussion section interpreted it (which is why the MTTs, catchment attributes, and uncertainties appear here). This is a common, albeit not universal, way of organising papers and is our preference. Perhaps the editor can comment as to their preferred structure for HESS.

As outlined above we significantly reordered the material to take into account the suggestions of both reviewers and the editor.

4. There are a lot of regressions were the only measurement for curve fit is the r2, using the p-value would also add information on the data that is correlated.

Agreed. P-values will be presented in the revised manuscript.  While these are useful, they do not change the overall conclusions.

We have included the p-values throughout the paper

Specific comments:

1. P1 L18: 2.4 to 3.2 TU is the annual average value, not the real range of activities, big difference, which might explain partially the low values obtained on the stream water.

As discussed above, there is little evidence for a strong seasonal variation of recharge in this catchment. For catchments with long MTTs, the annual variation in the $^{3}$H activities of rainfall has little impact where the MTTs are in excess of a few years. As also noted above, we will add a sentence or two to the discussion (Section 6.3) to explain this. The lowest $^{3}$H values of <1 TU are much lower than any recorded in rainfall (either annual averages or seasonal measurements) and consequently, the water must be at least several years old.

We have discussed this in more detail in section 5.3 (lines 463-474).

2. Page 6 Line 148: "agricultural" I think the authors meant "agriculture" or rephrase.

Correct "agricultural" should be "agriculture".  This will be corrected in the revised manuscript.

This has been corrected

3. P7 L169: Is the forest cover in Table 1 only eucalyptus? If so, reference table 1.

No, the catchment percentages of forest cover presented in Table 1 include native eucalyptus as well as production forestry (much of which is eucalyptus).  Here, we are providing a general description of the catchments but can add a few more words to clarify this.

We added this detail (lines 156-157).

4. P15 L390 to P16 L410: should be in "Study Area" section.

We disagree. Most of this material was derived as part of this study. While the WMIS website does provide estimates of catchment areas for most of the gauges, the values quoted here are from out GIS analyses. The other catchment attributes discussed here were calculated specifically for this

project. The Study Area (section 3) summarises material from previous studies rather than results that arose from this study.

Following the suggestion of the editor, we moved this material to the study area section (lines 203-209).

5. P16 L411-412: What about the correlation with geology? As well as a multiple regression or a PCA?

Noted. There is likely a correlation between runoff coefficient and geology (and/or possibly slope) for three of the catchments (Larnders Gauge, Upper Lardners, and James Access), as these catchments are relatively steep compared to the other catchments and are underlain almost entirely by the low-permeability Otway Group basement rocks. The variation in MTTs is probably controlled by multiple factors and while multiple regression analysis could be carried out, the small number of data points and the large number of potential controlling catchment attributes make it difficult to derive a unique solution (the same holds for other approaches such as PCA).

We noted the lack of correlation with geology (lines 514-517). Given the small number of samples multiple correlations and PCS are not warranted.

6. P16 L413-Fig7: This is a good example where the p-value could give more information, it's easy to see there are two extreme points with higher runoff coefficient that create that "correlation", but if those two would not be there the slope of the correlation would be negative instead of positive.

Agreed that including P-values would be useful. However, as noted above, the conclusions do not change.

We have added the p-values throughout

7. P17 L433-P18 L475: This should be in Results, not Discussion (with few exceptions of a couple of sentences that were discussion).

In the paper we have made the distinction between Results (which reports what is measured) and Discussion (the interpretation of the data). Thus, the $^3$H measurements are included in the results and the MTTs are included as discussion as these are the interpretation of the $^3$H data. This is our preferred discussion, although we acknowledge that there is no standard way of doing this (either in HESS or other journals). Perhaps the editor can best advise which structure is the best fit for HESS.

For the reasons outlined above we have kept this in the Discussion.

8. P18 L 476: The authors could make a section called uncertainties in the "Methodology" section with a description of each of them so there is no need to explain them in the "Discussion" section.

Noted. We included a brief description of uncertainties in MTT determination in the Introduction section (Line 108) as they are part of the background to understanding MTTs. This follows a similar format to other papers of ours and other authors. However, we can move this material to the Methodology section.

We moved this material to the Methods section as indicated

9. P19 Eq 2: this equation goes in the "Methodology" section, not discussion. Additionally, the "d" is missing in the equation, which is correctly mentioned in the text afterwards.

This would be better in the Methodology section. The "d" is subscript and this will be corrected in the revised manuscript.

We moved this material to the Methods section as indicated and corrected the equation (Eq. 2).

10. P20 L515-520: As mentioned before, 2.45 TU is probably the high end of activity on the annual precipitation.

There appears to be some confusion here as the $^3$H activity of 2.4 TU is on the **low**-end rather than the high-end of $^3$H activities of rainfall for this area (line 239). We agree that there is uncertainty in the $^3$H activity for modern rainfall.  This is why we re-calculated MTTs using a range of $^3$H activities (2.4 TU to 3.2 TU) which encompasses the range given by Tadros et al. (2014).  As we noted in the response to the other reviewer, the assumed $^3$H activities of modern rainfall make little difference to the calculated MTTs in waters with long MTTs such as these.

We have specified that the $^3$H activities are weighted mean annual values and that the value of 2.8 TU represents the most likely value but that the possible range may be between 2.4 and 3.2 (lines 300-309).

11. P20 L523: If 2.45 TU is on the high end, for the March calculations the precipitation should be more on the 1-1.3 TU (being conservative).

We are unsure where the $^3$H activities of 1-1.3 values come from. The measured $^3$H activity (2.45 TU) in the single precipitation sample that we collected is the lowest recorded $^3$H activity in rainfall for any area in Victoria, Australia (that we know of).  The measured average annual $^3$H activities (both from our studies and the IAEA datasets that Tadros et al., 2014 quote) are in the range 2.4 to 3.2 (lines 340 to 345). We will ensure that the $^3$H activities in rainfall are clearly explained in the revised manuscript.

As discussed above, we have clarified the rainfall input (lines 300-309).

12. P20 L530-531: Yes unimportant for the surveys taken in September, partially for those in November and July, I don't think it was unimportant in March.

The March samples have the lowest $^3$H activities, so the impact of the uncertainties in modern rainfall $^3$H activities will have the lowest relative impact on the estimated MTTs. This sentence could usefully be expanded to explain the relative impacts more clearly.

We expanded the discussion of uncertainties (Section 5.3) and have also specified which impact the understanding of relative MTTs within and between the catchments (lines 494-510).

13. P21 L539-L544: This should be in the Results section.

We disagree as this is part of the interpretation of the results. We will, however, reword this section so that it better conveys the point that we are interpreting data not presenting new data (e.g., "Given that the analytical uncertainty of the $^3$H are… …the resultant uncertainties in MTTs are…). The uncertainties were presented in the Methods (Section 4.2) and in Table 3 and we will refer to those sources here.

As well as the reordering of sections, we rephrased Section 5.3 to better convey that we are discussing the consequences of the results here rather than presenting new data.

14. P21 L547-549: I agree that the intermediate flow rates are important, maybe even the

This comment is incomplete, so we are not entirely sure of its meaning. It appears that you are agreeing with our conclusion that the greatest uncertainty in MTT estimates are for waters with intermediate $^3$H activities. We believe that this is an important conclusion.

15. P21 Eq 3: This equation belongs to "Methodology", not discussion.

Agreed, we will move it to the Methods section

We moved this as indicated (it is now Eq. 2).

16. P22 Eq 4 and 5: These should be in results.

Agreed, we will move them to the Methods section

This section was not included in the final version (as noted above).

17. P22 L569-582: This belongs to results.

This is a section that interprets the results, thus we consider that it is in the correct section

This section was not included in the final version (as noted above).

18. P22 L588-P23 L592: Discuss why there is no increase on the sulphate concentration in the Ten Mile Creek, are the anthropogenic activities different in this catchment than in the others?

We are unsure as to why there is no correlation between sulphate concentrations and discharge at Ten Mile Creek, when such correlations (including nitrate) do appear to exist at Upper Lardners and James Access. Sulphate concentrations are much higher at Ten Mile Creek than they are at Upper Lardners and James Access (Figure 6), which probably reflects the fact that Upper Lardners and the Gellibrand River at James Access are more pristine streams than Ten Mile Creek. Clearly, more data would help elucidate whether such correlations are real. We will touch upon this in greater detail within our revised text.

As noted above, we integrated this material into the other sections. Specifically, the observations that nitrate correlates with $^3$H and streamflow in in section 4.3 (where other correlations with major ion geochemistry are discussed) and the possible implications of this are on lines 582-584.

[revised manuscript text omitted]

---

## Author Response (AR2)

We thank the reviewer for these additional comments on the paper. We have addressed these below (responses in green) and have tried to incorporate or clarify the text in the paper.

General comments:

1. Throughout the manuscript there is an emphasis on using 2.4-3.2 as the TU for precipitation. However, in Tadros et al. (2014) these values are given as a rather coarse annual average activity values for the precipitation, which is very different to the intra-annual range of precipitation tritium activity. For example, in Tadros et al. (2014) they show the monthly variability in Alice Springs and in Brisbane, where the activity ranges were 7.5-20.6 TU and 3.6-8.2 TU respectively averaging all data collected for these two sites which include older data as well. The important point here is to emphasize that the high values were during autumn-winter in September, whereas the low activities were observed during the summer season (January to April). The high values were more than 2 times larger than the low activities. The 2.4-3.2 TU values are a good reference to have, but it is acceptable for this to be off the real values, considering the large area this map was covering, having to compromise in resolution. Summing up, the values 2.4-2.8 TU and 2.8-3.2 TU are not the annual range rather than just the annual average, missing the full range of the sampling. Additionally, observing the map with the precipitation 3H activities would also show that the values for Alice Springs and Brisbane are not covering the whole range they showed in the monthly values.

As we discuss below, we consider that the annual average of the precipitation is most relevant. As with the stable isotopes and major ions (e.g., Stewart et al., 2010), the long MTTs will result in the seasonal variation of $^3$H activities being smoothed out in the flow systems (Morgenstern et al., 2010). Our approach is similar to that used in most publications that use $^3$H to estimate MTT. Seasonality is important if recharge occurs just during one season, and we addressed that (lines 577-579).

While there may be local variations in the annual average $^3$H in rainfall, the use of a broad range of $^3$H activities amply accounts for that. The annual average $^3$H activities of 2.4 to 3.2 applies to modern rainfall over much of the centre and SE of Australia, including Alice Springs, Adelaide, and Melbourne, which is a much larger area than the study area. As outlined in the responses below, where we have measured multi-month $^3$H activities of rainfall (4 to 5 sites around Victoria) they are close to those predicted by Tadros et al. (2014). The annual average $^3$H activities in those studies are between 2.7 and 3.0 TU and again these samples were collected over a larger area than the study area. These sites include one near the coast (Melbourne) and the highlands of NE Victoria (Mount Buffalo). Perhaps our use of "range" is confusing. We were referring to the likely variability of the annual $^3$H averages, not the range in rainfall across the year; we have clarified that in the paper.

2. In this study the precipitation sampled was taken for 78 days during the period of the year of highest activity in the rain and it was 2.45 TU, leading to assume that the precipitation preceding the sampling of March 2015 could be ranging in activities close to half the sampled 2.45 TU, as discussed in the previous point. Since this study attempts to estimate the MTT at different times of the year, it is crucial to consider the correct end members, it is understood that the authors do not have such data but it is clear to me that the precipitation end-member used for March 2015 will not be the same as the precipitation for the other samplings. Even if the samplings are done during recession, the previous precipitations could have an influence that was not considered in this study because of the assumption of having such a high value on the precipitation activity. I do not believe this would change the overall conclusion of the study, but yes a significant change for the March 2015 sampling.

We agree that the [3]H activity of the 78 day precipitation sample is probably not representative of the annual rainfall, but we do not use it for that purpose. We estimate the average modern rainfall [3]H activity to be 2.8 TU. This is based on the Tadros et al., (2014) data and the [3]H activities of rainfall near Melbourne of 2.70 to 2.77 TU. However, in recognition of the uncertainty in rainfall [3]H activities we carried out the MTT calculations with modern rainfall [3]H activities between 2.4 to 3.2 (which as outlined above covers the estimated values for most of southeast Australia).

The geochemistry of the stream samples implies that the rivers are largely fed from water from within the catchment at the streamflows sampled in this study – this is probably especially the case in March at the end of the austral summer when streams in general in southeast Australia are fed by baseflow. Taking into account the seasonal (or shorter) variation of [3]H activities in rainfall would be relevant if there was direct contribution of recent rainfall into the streams, but that does not seem likely (especially during the summer low flows). As noted above, and as is now discussed in the paper (lines 570-579), the long MTTs mean that the seasonal variation of [3]H is less relevant than the annual average.

I see two possible scenarios regarding the activities in March 2015, the first one: that precipitation 3H activity is different in the northern catchments than on the southern catchments (since the rain gauge was in the northern region). That hypothesis is not possible to test with the available data, and maybe even not true, but I don't know if there is a special situation with the clouds on those catchments.

While we admit that our rainfall record is not complete, it is unlikely that there is a significant difference. The topography is not extreme and the high points are hills not mountains. As outlined elsewhere in the response, the [3]H activities of annual rainfall samples in SE Australia generally fall close to what is expected. The [3]H activities of annual rainfall collected from two sites separated by ~10 km in the Victorian Alps are 2.85 to 2.99 TU which agrees with the expected range of 2.8 to 3.2 TU from the Tadros et al. (2014) study. The [3]H activities of aggregated annual rainfall samples from two sites near Melbourne separated by ~30 km are 2.72 and 2.77 TU, which again are closely similar and within the expected range of values.

Nevertheless, we can ignore this scenario as it cannot be tested. Scenario 2: assuming precipitation 3H activity is homogeneous over all catchments, and knowing that the precipitation sampled and analyzed in this study was a weighted average of the rising activity limb; this would imply that there was rain with lower and higher activities than 2.45 TU. Therefore, you could assume that during summer season precipitation the 3h activity could have been on the range of 2 TU, in order to still be a valid end-member. I would consider this a conservative number, and better than testing towards 2.8 o 3.2 for this period of the year were rain 3H activity is lower rather than higher.

We discuss the issue of seasonality below. We agree if the rivers were fed by recent rainfall, we would need to take that into account, but especially in summer that seems to be unlikely.

I would like to finish with remarking that I really enjoyed reading and getting involved on the development of this study.

We certainly appreciate the time and effort and these comments illustrate some future directions that we could pursue to understand the input of water over flood peaks.

Specific comments:

1. Page 1 Line 8: it should say "rainfall average", as it is not the range.

We agree that it looked like a range and have rephrased it to make it clear. We checked the paper for consistency on this point and modified the wording throughout.

2. P6 L129: I think you meant to say "Identifying".

Corrected

3. P10 L228: It called my attention the second p-value, is that correct? 10^(-195)?

The datasets are large (several thousand points) and the p-values for the generally good $R^2$ values are correspondingly low. We have just stated here that all the p-values are $<10^{-6}$.

4. P10 L236: When in September was it sampled? September is the month with highest tritium activity, if the sampling included this month can give an idea on how high on the spectrum is this sample.

The sampling date 29th September (was in the Supplement Table, now referred to in the text). The response to Q5 addresses the seasonality issue.

5. P15 L354: "The lower than expected 3H… …representing rainfall of only part of the year". From the discussion above this is not a good reason, since the sampling was taken during the part of the year were precipitation activity is highest. This just tells that your range is lower than what you are assuming.

It is predicted that $^3H$ activities will be higher in the early spring months. However, data from Cartwright and Morgenstern (2015) show that a 3 month sample collected in mid-September in northeast Victoria had a lower $^3H$ activity (2.71 TU) than the annual average (2.85 to 2.99), which is a similar observation to that made in this paper. In that region, the measured annual $^3H$ rainfall activities are within the range predicted by Tadros et al., (2014) (between 2.8 and 3.2 TU). It is possible that the seasonal variation in individual years varies or that the $^3H$ activity of the rainfall in the months prior outweigh any higher $^3H$ activities from September rainfall.

This paragraph establishes the modern rainfall $^3H$ value and we have reworded it to make that clear. While there is some uncertainty, the estimates of Tadros that are based on multiple years of data over Australia as a whole are the most comprehensive. Our initial estimate of 2.8 TU is based on interpolation of those data and is similar to measured annual averages of 2.7 to 2.8 for the Melbourne area, which is ~150 km away and a similar distance from the coast. We used a wide range of values between 2.4 and 3.2 to assess the impact of the uncertainty in rainfall $^3H$ on the MTTs. This range easily encompasses all of our measured monthly to yearly rainfall $^3H$ activities from southeast Australia and actually encompasses the predicted $^3H$ activities in rainfall across all of central and southeast Australia, including Alice Springs, Adelaide, and Melbourne. We have thus been conservative in our assumptions.

One of the $^3H$ values was referred to as unpublished – this is now published (Cartwright et al., 2018) and we have updated the reference.

6. P17 L406-407: "Additionally the 3H activities plateau at ~2.0 TU, which is significantly lower than those of modern rainfall (Fig. 4)". Again, I disagree with how this is written, it is correct in some way, but not how it is written because the modern rainfall is not what you show in Fig. 4. I do believe that the activities in the streams always plateau at lower values than the precipitation; however, you have to treat each time of the year as separate cases. March 2015 3H activities plateau at X TU, which is lower than the perhaps ~1 TU on precipitation at this time of the year, whereas the samplings taken during autumn plateau at ~2.0 TU, which is lower than the modern precipitation at

this time of the year. Though somehow, Gellibrand River at JA and both Lardners have a larger than 1 TU at the March sampling, maybe something that differentiates them from the other catchments?

This sentence was a description of the data that was probably out of place here as it simply repeated the observations in Section 4.2. We agree that it was ambiguous as written and have clarified that we are making the comparison with annual average $^3$H activities.

Although the IAEA dataset is not very complete, there is no evidence that rainfall in March has a $^3$H activity of as low 1 TU. Rather, as reported elsewhere in the paper (lines 579-589), the $^3$H activities in summer rainfall are expected to be closely similar to the average annual $^3$H activities. An observation that the $^3$H activities of summer (December to February) rainfall at Mount Buffalo in northeast Victoria were similar (2.86 TU) to those of two annual rainfall samples (2.99 and 2.85 TU) support that assertion (data from Cartwright and Morgenstern, 2015, reported on lines 584-587).

The seasonal variation in $^3$H is important where streams are fed by recent rainfall, however, the general understanding of the hydrogeology of southeast Australia is that streamflow is sustained by baseflow during the dryer summer months. The observations outlined on lines 484-502 also imply that there is little direct input of rainfall into the streams at the streamflows represented in this study. Our aims in this study were to characterise MTTs during the "average" flow conditions in the different seasons and we deliberately avoiding sampling over the high flow events triggered by rainfall (in those cases the $^3$H activities of the event water would be important). Where transit times are in excess of a few years, one would not see any seasonal signal preserved in the catchment waters (now noted on lines 577-579) and the main impact of seasonality would be if preferential recharge had occurred (we discuss seasonal recharge on lines 578-589).

7. P19 L474: I think you meant Eq. (2).

Corrected

8. P20 L485: Again Eq. (2).

Corrected

9. P21 L520: "…estimate likely controls the MTTs." I think it should be control instead of controls.

This was a convoluted sentence and we have rewritten it.

10. P22 Section 5.5: This paragraph shows results rather than analysis. Placed where it is it looks more like a "fun fact", since it not used later and it does not add to the overall conclusion of the paper.

We disagree that these are results and the calculation uses the Mean Transit Times and needs to follow those. However, it is correct that the paragraph was isolated where it was and probably does not warrant a section of its own. We have moved it to the end of section 5.2 where we discuss the Mean Transit Times to emphasise the connection.

11. P22 L559: "…between catchments in important…" I think you meant to say is.

Corrected

12. Fig 4: Text 'in Figure' says "Expected range of 3H…", change to expected annual average, since it is not the range.

Changed to be consistent with how we express this in the text.

13. Fig 7 was called in the manuscript after figures 8, 9 and 10. Change the order or call it earlier.

We have renumbered the figures to reflect the order in which they are referred to (Fig.7 is now Fig. 10).

14. P-values were stated throughout the text but were missing in all figures, I would like to see the value in the figure as well, rather than having to look for it in the text when I am studying a figure.

We have added these

[revised manuscript text omitted]